# Coupling of pupil- and neuronal population dynamics reveals diverse influences of arousal on cortical processing

**Thomas Pfeffer**[1,2]*†, **Christian Keitel**[3,4]†, **Daniel S Kluger**[5,6], **Anne Keitel**[7], **Alena Russmann**[2], **Gregor Thut**[4], **Tobias H Donner**[2‡], **Joachim Gross**[4,5,6‡]

[1]Universitat Pompeu Fabra, Center for Brain and Cognition, Computational Neuroscience Group, Barcelona, Spain; [2]University Medical Center Hamburg-Eppendorf, Department of Neurophysiology and Pathophysiology, Hamburg, Germany; [3]University of Stirling, Psychology, Stirling, United Kingdom; [4]Centre for Cognitive Neuroimaging, Institute of Neuroscience and Psychology, University of Glasgow, Glasgow, United Kingdom; [5]Institute for Biomagnetism and Biosignal Analysis, University of Münster, Malmedyweg, Muenster, Germany; [6]Otto Creutzfeldt Center for Cognitive and Behavioral Neuroscience, University of Münster, Muenster, Germany; [7]University of Dundee, Psychology, Dundee, United Kingdom

**\*For correspondence:**
thms.pfffr@gmail.com

†These authors contributed equally to this work
‡These authors also contributed equally to this work

**Abstract** Fluctuations in arousal, controlled by subcortical neuromodulatory systems, continuously shape cortical state, with profound consequences for information processing. Yet, how arousal signals influence cortical population activity in detail has so far only been characterized for a few selected brain regions. Traditional accounts conceptualize arousal as a homogeneous modulator of neural population activity across the cerebral cortex. Recent insights, however, point to a higher specificity of arousal effects on different components of neural activity and across cortical regions. Here, we provide a comprehensive account of the relationships between fluctuations in arousal and neuronal population activity across the human brain. Exploiting the established link between pupil size and central arousal systems, we performed concurrent magnetoencephalographic (MEG) and pupillographic recordings in a large number of participants, pooled across three laboratories. We found a cascade of effects relative to the peak timing of spontaneous pupil dilations: Decreases in low-frequency (2–8 Hz) activity in temporal and lateral frontal cortex, followed by increased high-frequency (>64 Hz) activity in mid-frontal regions, followed by monotonic and inverted U relationships with intermediate frequency-range activity (8–32 Hz) in occipito-parietal regions. Pupil-linked arousal also coincided with widespread changes in the structure of the aperiodic component of cortical population activity, indicative of changes in the excitation-inhibition balance in underlying microcircuits. Our results provide a novel basis for studying the arousal modulation of cognitive computations in cortical circuits.

## Editor's evaluation

The authors have mapped how fluctuations of pupil-linked arousal and cortical activity co-vary in the human brain at rest recorded using MEG. This was achieved in a large sample of participants (N=81), and the results reveal diverse and consistent arousal effects on band-limited cortical activity. These findings provide important insight into how subcortical activity associated with arousal is reflected in neocortical dynamics.

## Introduction

Variations in cortical state profoundly shape information processing and, thus, cognition (*Busse et al., 2017*; *Fu et al., 2014*; *Zagha et al., 2013*). These variations occur continuously due to subtle fluctuations in the level of arousal, and even in the absence of changes in overt behavior (*Harris and Thiele, 2011*; *McGinley et al., 2015b*). Two key mediators of arousal-dependent variations are the brainstem nucleus locus coeruleus (LC), which supplies noradrenaline (NE), and the basal forebrain (BF), which supplies acetylcholine (ACh) (*Harris and Thiele, 2011*; *Hasselmo, 1995*; *Lee and Dan, 2012*; *Steriade, 1996*). It has long been thought that these nuclei are organized homogeneously and innervate cortical target regions diffusely, and indiscriminately across cortical regions. This has led to the idea that the arousal system acts as a global 'broadcast signal' and uniform controller of cortical state (*Aston-Jones and Cohen, 2005*; *Harris and Thiele, 2011*; *Leopold et al., 2003*; *Turchi et al., 2018*).

More recently, however, this view has been challenged: Several neuronal subpopulations with distinct projection targets have been found in both the LC and BF (*Chandler et al., 2013*; *Chandler et al., 2019*; *Sarter et al., 2009*; *Schwarz and Luo, 2015*; *Totah et al., 2018*; *Zaborszky et al., 2015*; *Záborszky et al., 2018*). In addition, neuromodulator receptors exhibit a rich and diverse distribution across the cortex (*Burt et al., 2018*; *van den Brink et al., 2019*). In this report, we comprehensively show how this structural and molecular heterogeneity translates into a multitude of arousal effects on neuronal population activity across the human cortex.

We used pupil diameter as a proxy for arousal (*Beatty, 1982*; *Bradshaw, 1967*; *Hess and Polt, 1964*; *Kahneman et al., 1967*; *Kahneman and Beatty, 1966*). Pupil diameter has recently attracted interest in neuroscience as a peripheral index of how arousal influences cortical state (*McGinley et al., 2015b*; *Reimer et al., 2014*; *Vinck et al., 2015*). Spontaneous changes in pupil diameter mimic the effects seen during the behavioral transition from quiet wakefulness to locomotion in rodents (*Crochet and Petersen, 2006*; *Niell and Stryker, 2010*; *Polack et al., 2013*; *Poulet and Petersen, 2008*): suppression of low (<10 Hz) and elevation of higher frequency (>30 Hz) activity (*McGinley et al., 2015b*; *Reimer et al., 2014*; *Vinck et al., 2015*). Critically, fluctuations in pupil diameter under constant retinal illumination track LC-NE activity (*Breton-Provencher and Sur, 2019*; *de Gee et al., 2017*; *Joshi et al., 2016*; *Murphy et al., 2014*; *Reimer et al., 2016*) as well as BF-ACh activity (*Reimer et al., 2016*). Fluctuations in pupil diameter are reflected more strongly in cholinergic activity, whereas

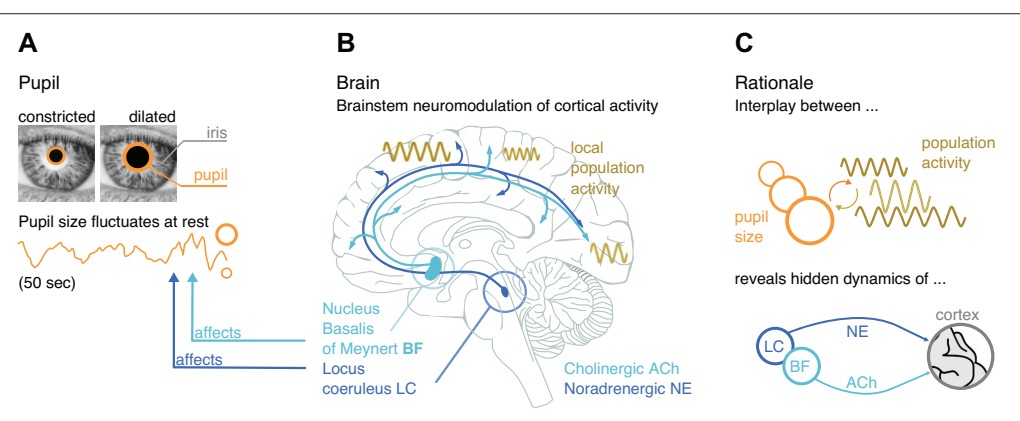

**Figure 1.** Rationale of our approach and summary results. (**A**) Even at rest and under constant illumination, pupil size (diameter) fluctuates spontaneously. (**B**) The locus coeruleus (LC) in the brainstem and the nucleus basalis of Meynert in the basal forebrain (BF) affect pupil size. LC releases the neuromodulator noradrenaline (NE) to the cortex, BF releases acetylcholine (ACh). Synchronous local neuronal population activity across the brain can be measured with magnetoencephalography (MEG). (**C**) Rationale: Studying the coupling of pupil size and local population activity recorded with MEG provides insights into the functional influences of deep-brain structures in brainstem (LC) and BF. Panel B modified from *van den Brink et al., 2019*.

The online version of this article includes the following figure supplement(s) for figure 1:

**Figure supplement 1.** MEG power spectra across recording sites.

**Figure supplement 2.** Power spectra and signal components of pupil time series, for all magnetoencephalographic (MEG) laboratories.

the rate of change in pupil diameter (i.e., its temporal derivative) is most strongly associated with noradrenergic activity (*Reimer et al., 2016*).

Animal research has mostly been limited to circumscribed sensory cortical areas in rodents (*McGinley et al., 2015a*; *Reimer et al., 2014*; but see *Shimaoka et al., 2018*) and in the anterior cingulate cortex of the macaque brain (*Joshi and Gold, 2020*). Insights from non-invasive magnetic resonance imaging (MRI) in humans, on the other hand, have revealed spatially rich correlations of cortical activity with pupil-linked arousal (*Groot et al., 2021*; *Schneider et al., 2016*; *Yellin et al., 2015*). However, the low temporal resolution of these measurements complicates direct comparisons with results from electrophysiological recordings in rodents and non-human primates. Therefore, we still lack a comprehensive understanding of which features of cortical population activity are shaped by arousal and how such effects are distributed across cortex.

To close these gaps, we assessed the covariation of spontaneous fluctuations in pupil-linked arousal and whole-head magnetoencephalographic (MEG) recordings in resting human participants across brain regions (*Figure 1*), components of cortical activity (oscillatory and aperiodic), and temporal lags. Our findings provide a comprehensive picture of relationships between intrinsic fluctuations in pupil-linked arousal and neural population activity in different brain regions. First, we demonstrate opposite effects of pupil-linked arousal on low- and high-frequency components of cortical population activity across large parts of the brain, similar to previously observed effects in rodent sensory cortices. These distinct effects occurred in a systematic temporal sequence. Second, we also uncovered hitherto unknown, nonlinear (inverted U-shaped) arousal effects in occipito-parietal cortex. Third, we identified spatially widespread correlations between pupil-linked arousal and the structure of aperiodic activity, suggesting underlying changes in cortical excitation-inhibition balance (*Gao et al., 2017*; *Waschke et al., 2021*).

## Results

We analyzed data from concurrent whole-head MEG and pupil recordings in 81 human participants, collected at three MEG laboratories (Glasgow, UK; Hamburg & Münster, Germany). All recordings were carried out under constant illumination and in sound-attenuated environments, thus removing fluctuations in pupil size due to changes in sensory input. Participants were instructed to keep their eyes open, while fixating and resting otherwise. The length of individual recordings varied between 5 and 10 min (see Materials and methods for further details).

For each of the three MEG laboratories, we obtained time courses of MEG activity and of pupil diameter fluctuations. A first control analysis of the recordings (*Figure 1—figure supplements 1 and 2*) established that they were highly comparable between laboratories, despite the differences in locations, setups, and participants.

In what follows, we refer to the time course of pupil diameter as 'pupil'. Previous work in rodents suggests that this time course and its temporal derivative indicate distinct (cholinergic and noradrenergic, respectively) neuromodulatory inputs to cortex (*Reimer et al., 2016*). Thus, we also tested for relationships of MEG activity with the first temporal derivative of the pupil diameter time course (hereafter: 'pupil derivative'). We found that, in our data, the results were largely similar for pupil and pupil derivative. Therefore, the main figures show only results for pupil, whereas the results for pupil derivative are presented in the Supplement.

### Robust pupil-brain couplings across the spectrum of cortical population activity

We evaluated the strength of pupil-brain coupling across a wide range of frequency bands in MEG sensor space by means of mutual information (MI), which is sensitive to monotonic as well as non-monotonic associations between two signals (*Ince et al., 2017*). We calculated the MI between pupil traces and power time courses of band-limited cortical activity across 25 log-spaced center frequencies from 2 to 128 Hz at each MEG sensor (see *Figure 1—figure supplements 1 and 2* for power spectra of MEG and pupil size time courses for each laboratory). These MI values were standardized against permutation distributions of surrogate MI values obtained by time-shifting the pupil time series against the power time courses, then averaged across all sensors for a first summary statistic, separately for each laboratory (see Materials and methods).

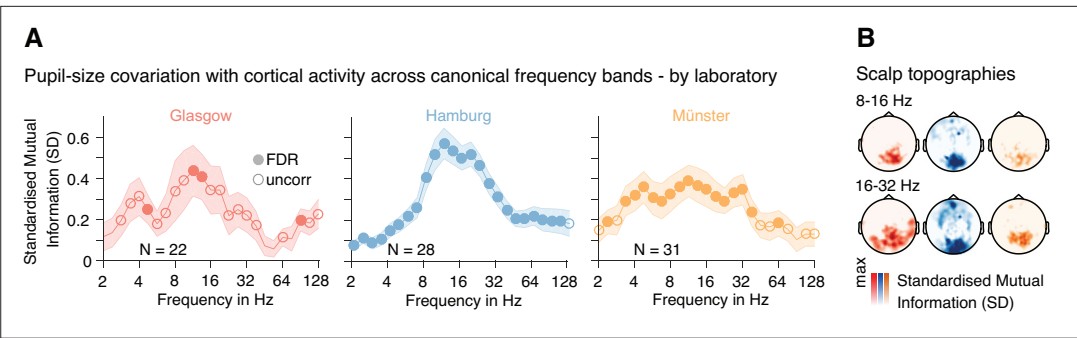

**Figure 2.** Mutual information between MEG and pupil time series. (**A**) Mutual information (MI) spectra demonstrate the covariation between pupil size and power fluctuations across canonical frequency bands from 2 to 128 Hz (note the log-spaced frequency axis). MI values are standardized against individual permutation distributions based on a temporal re-shuffling of pupil size and power time courses (for details, see Materials and methods). Filled/open markers in plots signify frequency-specific t-tests of individual standardized MI values against zero (two-tailed) at FDR-corrected thresholds ($\alpha_{Glasgow}$ = 0.002; $\alpha_{Hamburg}$ = 0.008; $\alpha_{Münster}$ = 0.003), and an uncorrected threshold of p < 0.05. Data pooled across all magnetoencephalographic (MEG) sensors. Shaded areas depict the standard error of the mean across subjects. Color-coding refers to the recording site; left: Glasgow data in red; center: Hamburg data in blue; right: Münster data in yellow. *N* denotes the number of samples recorded at each site. (**B**) Scalp topographies illustrate the commonalities of the spatial distribution of pupil size power covariations for two prominent frequency ranges. As in A, colors code the recording site – from left to right: Glasgow, Hamburg, Münster.

The online version of this article includes the following figure supplement(s) for figure 2:

**Figure supplement 1.** Analyses of microsaccade-locked cortical activity and pupil diameter.

For all three MEG laboratories, standardized-MI spectra showed similar patterns of widespread pupil-brain associations across frequencies with prominent peaks in the 8–32 Hz range (*Figure 2A*). MI scalp distributions within this frequency range further underpinned the commonalities between recording sites despite different MEG systems and setups (*Figure 2B*). Significant relationships with pupil size were also evident for the low-frequency (4–8 Hz) and high-frequency (64–128 Hz) components of MEG power (*Figure 2A*) in all three datasets.

A possible confound in identifying relationships between pupil and cortical dynamics are eye movements: Saccades change pupil size (*Mathôt, 2018*; *Mathôt et al., 2015*) and oculomotor behavior is functionally linked to cortical alpha oscillations (*Popov et al., 2021*). Here, we controlled for saccade effects by regressing out canonical responses from pupil time series (as detailed in the Materials and methods section; see, e.g., *Urai et al., 2017*). An additional control analysis found that remaining microsaccades (not captured by the regressing-out) led to a transient suppression of cortical activity in the 8–32 Hz range. However, this effect was not associated with any changes in pupil diameter (*Figure 2—figure supplement 1*).

## Fluctuations in pupil diameter lag behind changes in cortical population activity in a frequency-dependent manner

Having established robust pupil-brain couplings, we quantified the temporal relationship between both signals. Changes of neural activity in brainstem arousal centers such as the LC generate changes in neural activity that exhibit considerable temporal lag (~500 ms in macaques) and temporal low-pass characteristics (*Breton-Provencher and Sur, 2019*; *de Gee et al., 2017*; *Hoeks and Levelt, 1993*; *Joshi et al., 2016*; *Korn and Bach, 2016*). This temporal smoothing is produced by the peripheral pathways (nerves and muscles) controlling pupil motility (*Hoeks and Levelt, 1993*; *Korn and Bach, 2016*). Thus, one might expect the effect of neuromodulatory arousal signals to occur earlier in cortical population activity (a single synapse from LC to cortex, unless mediated by a third structure, such as the thalamus) than in pupil diameter.

We computed cross-correlations between fluctuations in frequency-resolved sensor-level power and pupil size (Materials and methods). We found the maximum correlation magnitude between pupil diameter and power fluctuations at a frequency-dependent lag (*Figure 3A and B*): For activity

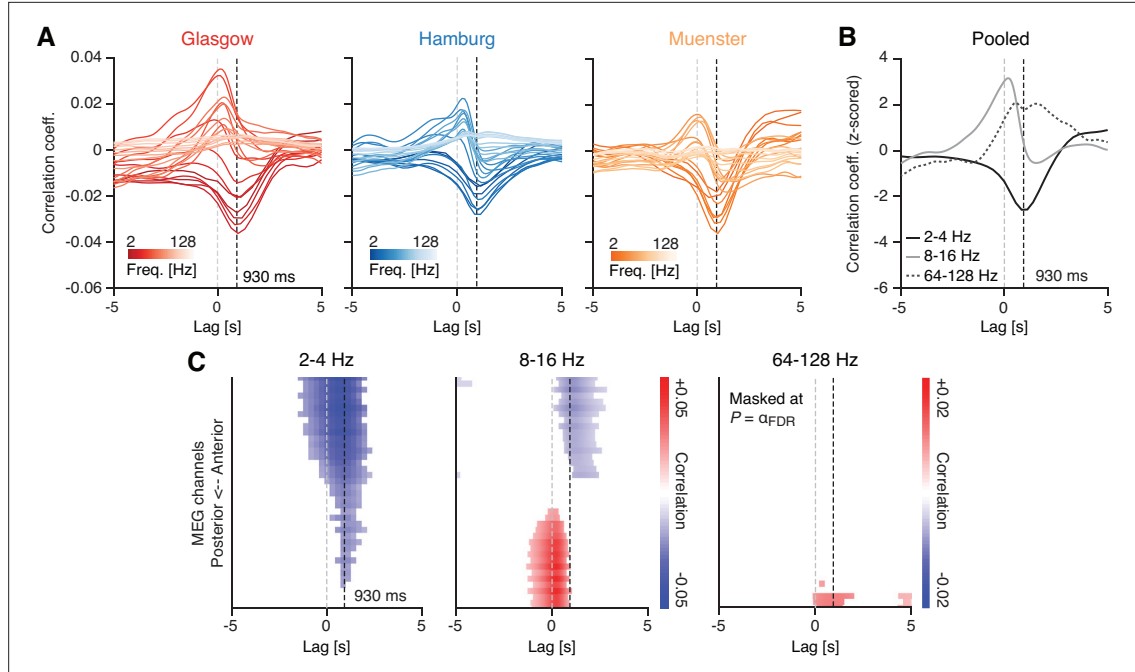

**Figure 3.** Sensor-level cross-correlations between fluctuations in pupil diameter and band-limited power fluctuations. (**A**) Correlation between fluctuations in pupil diameter and band-limited power fluctuations across lags, ranging from –5 to 5 s and for all three recording sites (left: Glasgow; center: Hamburg; right: Muenster). Colors indicate the magnetoencephalographic (MEG) frequency band, with darker colors representing lower frequencies and lighter colors representing higher frequencies. Negative lags are indicative of 'pupil preceding MEG', whereas positive lags are indicative of the opposite. The vertical dashed black line denotes a lag of 930 ms (**Hoeks and Levelt, 1993**) and the dashed light gray line a lag of 0 ms. (**B**) Same as in (**A**), but averaged across the three recording sites and for three frequency bands of interest: 2–4 Hz (black), 8–16 Hz (dark gray), and 64–128 Hz (gray dashed). (**C**) Correlation values across all channel groups, sorted from anterior to posterior, and for lags ranging from –5 to +5 s. The dashed gray line depicts a lag of 0 ms and the dashed black line shows a lag of 930 ms (**Hoeks and Levelt, 1993**). Correlation values were averaged within three frequency bands of interest: 2–4 Hz (left), 8–16 Hz (center), and 64–128 Hz (right). Masked at the corresponding FDR-adjusted significance thresholds (q = 0.1): $\alpha$ = 0.00122 (2–4 Hz), $\alpha$ = 0.0029 (8–16 Hz), and $\alpha$ = 0.0001 (64–128 Hz). p-Values were obtained from a two-tailed paired t-test.

The online version of this article includes the following figure supplement(s) for figure 3:

**Figure supplement 1.** Sensor-level cross-correlations between the temporal derivative of fluctuations in pupil diameter (pupil derivative) and band-limited power fluctuations.

**Figure supplement 2.** Spatial distribution of cross-correlations, separately for each magnetoencephalographic (MEG) site.

in the low- (around 2–4 Hz) and high-frequency bands (around 64–128 Hz), correlation magnitudes peaked at lags of 832 ms (negative peak) and 562 ms (positive peak), respectively. This result was highly consistent across the three MEG sites (**Figure 3A**). These latencies indicated power fluctuations that preceded corresponding variations in pupil diameter. Correlations with power fluctuations in the 8–16 Hz range, on the other hand, peaked at a lag closer to zero (210 ms; positive peak), indicating a temporally closer relationship (**Figure 3A and B**; see Discussion).

Fluctuations in pupil, and pupil derivative, may primarily reflect cholinergic and noradrenergic activity, respectively (**Reimer et al., 2016**). Consistent for all three datasets, the cross-correlation analysis for pupil derivative revealed peaks closer to zero lag for low and high frequencies (2–4 Hz negative peak at 277 ms, 64–128 Hz positive peak at –90 ms; **Figure 3—figure supplement 1**). In contrast, correlations with activity in the 8–16 Hz range peaked somewhat later (700 ms; negative peak; **Figure 3—figure supplement 1**). The shorter lags for the pupil derivative are consistent with previous findings in rodents (**Reimer et al., 2016**). It should be noted that, due to the nature of the employed wavelet analysis, the temporal resolution of the spectral estimates varies across frequency bands, which poses a potential challenge when comparing peak latencies across frequencies. In the present case, however, the differences in peak latencies between frequency bands were sufficiently large in order to be interpretable despite this limitation.

The cortical innervation profiles of subcortical arousal centers and the cortical receptor composition for the neuromodulators released by these centers exhibit substantial heterogeneity across cortex (*Ramos and Arnsten, 2007*; *van den Brink et al., 2018*; *van den Brink et al., 2019*). Therefore, we reasoned that the nature (frequency dependence and/or temporal lag) of pupil-brain couplings may differ considerably across cortical areas and tested the topographical profile of the cross-correlations. To account for the different channel layouts of the three MEG systems, we averaged cross-correlations, computed for individual MEG channels, within 39 non-overlapping channel groups, consisting of channels that were grouped based on their location along the anterior-to-posterior axis (see Materials and methods). In keeping with the previous analyses, we collapsed the data into three non-overlapping frequency bands of interest: 2–4, 8–16, and 64–128 Hz.

Consistent with the results obtained from sensor averages, a negative correlation between pupil diameter fluctuations and power in the low frequencies (2–4 Hz) peaked at a lag of around 900 ms, with a spatial profile that was relatively uniform across brain areas (*Figure 3C*, left). For the high frequencies (64–128 Hz), on the other hand, the positive peak was confined to posterior regions, with a lag similar to the low-frequency band (*Figure 3C*, right). In the 8–16 Hz-range, the picture was more nuanced (*Figure 3C*, center): The close-to zero-lag positive correlation dominated posterior sensors whereas anterior sensors registered a negative correlation with a peak around 1 s. This suggests that the correlation between pupil diameter and power in the 8–16 Hz range may have been driven by independent mechanisms that operate on distinct time scales (see Discussion).

The spatio-temporal correlation patterns between pupil size (and pupil derivative) and band-limited power separated by MEG recording site are shown in *Figure 3—figure supplement 2*.

In summary, we identified lags between spontaneous fluctuations in pupil diameter and low- and high-frequency activity that are consistent with previously reported lags between noradrenergic and cholinergic activity, respectively, and spontaneous pupil diameter fluctuations in rodents. In addition, we found that pupil-brain coupling in the intermediate 8–16 Hz range peaked with a time lag closer to maximum pupil dilation for posterior sensors. Our findings suggest that the correlations in different frequency bands, and for different cortices within the intermediate frequency range, may be mediated by different neuromodulatory systems (see Discussion).

## Spatial and spectral dissociations of pupil-power correlations

Having established the distinct temporal coupling profiles for different frequency bands, we now tested which cortical areas received the strongest modulation. Sensor space analyses suggested that effects of pupil-brain coupling were not uniformly distributed along the anterior-posterior axis (*Figure 3*). To map out cortical loci of frequency-specific pupil-brain coupling in detail, we used spectrally resolved source-reconstructed MEG data. Given the high consistency of the sensor space results, we report data pooled across the three recording sites for the following analyses.

Based on previous reports (*Hoeks and Levelt, 1993*; *Joshi et al., 2016*; *Reimer et al., 2016*), we shifted the pupil signal 930 ms forward (with respect to the MEG signal) for this analysis. We introduced this shift to compensate for the lag that had previously been observed between external manipulations of arousal (*Burlingham et al., 2021*; *Hoeks and Levelt, 1993*; *Wierda et al., 2012*) as well as spontaneous noradrenergic activity (*Reimer et al., 2016*) and changes in pupil diameter. In our data, this shift also aligned with the lags for low- and high-frequency extrema in the cross-correlation analysis (*Figure 3B*).

Using the forward-shifted pupil signal, we computed correlations with fluctuations in spectral power (*Figure 4*), separately for 8799 source locations covering the 246 regions of the Brainnetome atlas (*Fan et al., 2016*; see Methods). Averaged across all source locations, correlations between intrinsic pupil size and power fluctuations were frequency-dependent (*Figure 4A*): Consistent with findings from invasive recordings in primary visual cortex of rodents (*Reimer et al., 2014*; *Vinck et al., 2015*), we found a robust negative correlation between pupil diameter and power in the low frequencies (2–8 Hz) and a numerically small, albeit statistically robust, positive correlation in the high frequencies (between 50 and 100 Hz). To investigate the spatial distribution of the correlations, we sorted the regions of the Brainnetome atlas along the anterior-posterior axis. Whereas the correlations in the low frequencies covered large parts of the brain (except for a few posterior regions, *Figure 4B*), pupil-brain coupling in other frequency ranges exhibited a spatially richer structure, with positive (8–32 Hz) and negative (>50 Hz) correlations in more occipital regions and correlations of opposite sign in more anterior sources (*Figure 4B*).

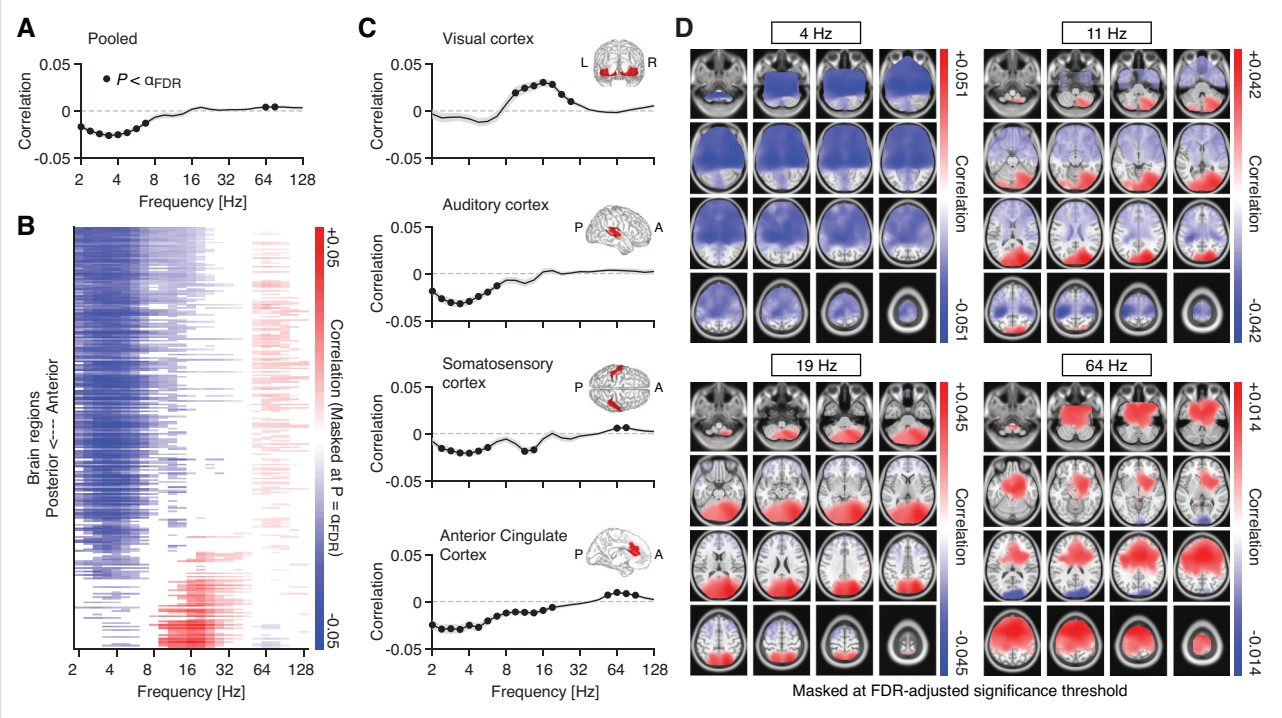

**Figure 4.** Source space pupil-power correlations. (**A**) Spectrum of pupil-power correlations, averaged across all 8799 source locations. Filled black circles denote significant correlations after FDR adjustment (two-tailed paired t-test; FDR-adjusted α = 0.008; with q = 0.1). The gray shaded area depicts the standard error of the mean across subjects. (**B**) Pupil-power correlations across the 246 regions of the Brainnetome atlas and sorted from anterior to posterior regions. Masked at FDR-adjusted significance threshold (two-tailed paired t-test; FDR-adjusted α = 0.0053; with q = 0.1). (**C**) Pupil-power correlations across four selected regions of interest (see Materials and methods for details; filled black circles denote significant correlations after FDR adjustment with q = 0.1): visual cortex (α = 0.007), auditory cortex (α = 0.007), somatosensory cortex (α = 0.011), and anterior cingulate cortex (α = 0.018; from top to bottom). Insets depict the spatial extent of the respective region of interest. (**D**) Spatial distribution of pupil-power correlations, across four frequency bands of interest. Correlations between pupil and power (in clockwise direction, starting from the top left) at center frequencies 4, 11.3, 19.0, and 64 Hz. All maps are masked at the FDR-adjusted significance thresholds with q = 0.1 (4 Hz: α = 0.0086; 11.3 Hz: α = 0.0069; 19 Hz: α = 0.0044; 64 Hz: α = 0.0039; all p-values were obtained from a two-tailed paired t-test).

The online version of this article includes the following figure supplement(s) for figure 4:

**Figure supplement 1.** Source space pupil-derivative-power correlations.

**Figure supplement 2.** Difference between pupil-derivative-power (*Figure 4—figure supplement 1*) and pupil-power correlations (*Figure 4*).

Previous studies have reported largely similar effects of pupil-linked arousal on population activity in the different sensory cortices (*McGinley et al., 2015a*; *Reimer et al., 2014*; *Vinck et al., 2015*; but see *Shimaoka et al., 2018*). Capitalizing on our source reconstructions, we tested if the correlations between pupil diameter and cortical activity varied across cortical regions of interest (see inset maps in *Figure 4C*), including three sensory areas (visual cortex, auditory cortex, and somatosensory cortex) as well as anterior cingulate cortex, a region that receives strong innervations from the LC (*Chandler et al., 2013*), while also projecting to the LC itself (*Aston-Jones and Cohen, 2005*; *Joshi et al., 2016*). Patterns of pupil-power correlations were highly region-specific: Correlations in the primary visual cortex were positive within the 8–32 Hz range (*Figure 4C*, top). Correlations in the auditory cortex, on the other hand, were dominated by negative correlations in the low-frequency range (<8 Hz; *Figure 4C*, second to top). Somatosensory cortex exhibited negative correlations in the low and intermediate frequencies (2–16 Hz) and, at the same time, positive correlations in the high-frequency range (~64 Hz; *Figure 4C*, second to bottom). Correlations with activity in anterior cingulate were significant across most frequencies, with a spectral pattern similar to auditory cortex (*Figure 4C*, bottom).

For further insights into the spatial distribution of the observed correlations, we mapped correlations of pupil and pupil-derivative time courses with band-limited power fluctuations across all 8799 source locations, again estimated at a lag of 930 ms for pupil and a lag of 0 ms for pupil derivative.

To this end, we focused on four frequency bands of interest, with the following center frequencies (*Figure 4D*): 4, 11.3, 19, and 64 Hz, corresponding to the classical theta, alpha, beta, and gamma frequency ranges. In the low frequencies (centered at 4 Hz), correlations were negative in the majority of source locations, with bilateral peaks in the anterior sections of the hippocampus (*Figure 4D*, top left). Significant positive correlations between pupil and activity in the alpha/beta range (11.3 and 19 Hz) were located in the visual cortices of both hemispheres (*Figure 4D*, top right and bottom left). Significant negative correlations were found in left and right motor and somatosensory cortices (*Figure 4D*, top right), although limited to activity in the alpha range (centered at 11.3 Hz). Similarly, correlations with high-frequency activity (centered at 64 Hz) were non-uniform across space, with negative correlations located in visual cortices bilaterally, and positive correlations across large parts of the prefrontal cortex (*Figure 4D*, bottom right).

Repeating the analyses for the pupil derivative, we found largely similar spatial patterns for the correlations, albeit with reduced magnitudes (*Figure 4—figure supplement 1*; see *Figure 4—figure supplement 2* for statistical comparison of pupil-power and pupil-derivative-power correlations). *McGinley et al., 2015a*, reported a similar reduction when comparing pupil derivative with pupil-based correlations in rodents.

Briefly summarized, source localization of pupil-brain coupling revealed frequency band-specific patterns of arousal influences on cortical processing. While coupling with low-frequency activity was relatively uniform, coupling with intermediate- and high-frequency activity showed effects of opposite signs for different cortices, even within frequency bands, indicating that arousal has discernible effects on different sensory cortices, as well as higher-order associative regions of the human brain at rest.

## Nonlinear relations between pupil-linked arousal and band-limited cortical activity

Above-described analyses focused on quantifying monotonic relationships between intrinsic fluctuations in pupil diameter, as well as pupil-derivative, and band-limited cortical population activity. However, arousal may exert more complex influences on cortical activity that roughly follow a nonlinear, inverted U shape, known as the Yerkes-Dodson law (*Yerkes and Dodson, 1908*; *Aston-Jones and Cohen, 2005*; *McGinley et al., 2015b*). Evidence suggests that gain of neuronal input-output functions as well as behavioral performance peak at intermediate arousal levels (*He and Zempel, 2013*; *McGinley et al., 2015b*; *Waschke et al., 2019*), flanked by lower gain and performance at lower (e.g., disengagement and drowsiness) and higher arousal (e.g., stress). We, therefore, reasoned that local pupil-power couplings may exhibit similar inverted U relationships. To test this, we divided the data into non-overlapping temporal segments of 2 s length and computed power spectra as well as mean pupil size (or mean pupil derivative) for each segment. Next, we averaged the power spectra within each of 14 evenly sized pupil bins, ranging from smallest to largest pupil diameter on each block. In keeping with previous analyses, we shifted the pupil time series forward by 930 ms, while applying no shift to the pupil derivative.

Focusing on three predefined frequency bands, the relation between pupil diameter and pupil derivative with low-frequency (2–4 Hz) and high-frequency activity (64–128 Hz) was well characterized by a monotonic relationship (*Figure 5A*, left and right) when averaged across all source locations. In contrast, activity in the 8–16 Hz range exhibited a high degree of nonlinearity that followed an inverted U relationship (*Figure 5A*, center).

To test for potential quadratic relationships exhaustively across frequencies and space, we fitted a second-order polynomial to the power-by-pupil bin functions, separately for each subject, recording block, source region and frequency (see Materials and methods). Next, we mapped the mean coefficient of the quadratic term of the polynomial ($\beta_2$) across space and frequencies. To account for differences in absolute power (offset), we normalized (i.e., z-scored) the power-by-pupil functions prior to model fitting. The analysis revealed that inverted U relationships were exclusive to the range of 8–32 Hz (i.e., $\beta_2$ significantly smaller than zero; two-tailed paired t-test; *Figure 5B*).

Focusing on the quadratic nonlinearity in four regions of interest (as used in the previous analyses) pointed toward locally emphasized intermediate-frequency inverted U shaped relationships (auditory and somatosensory cortices, *Figure 5C*). A spatially more fine-grained analysis of these effects revealed that the inverted U relationship in the alpha band (11 Hz) peaked in the superior parietal

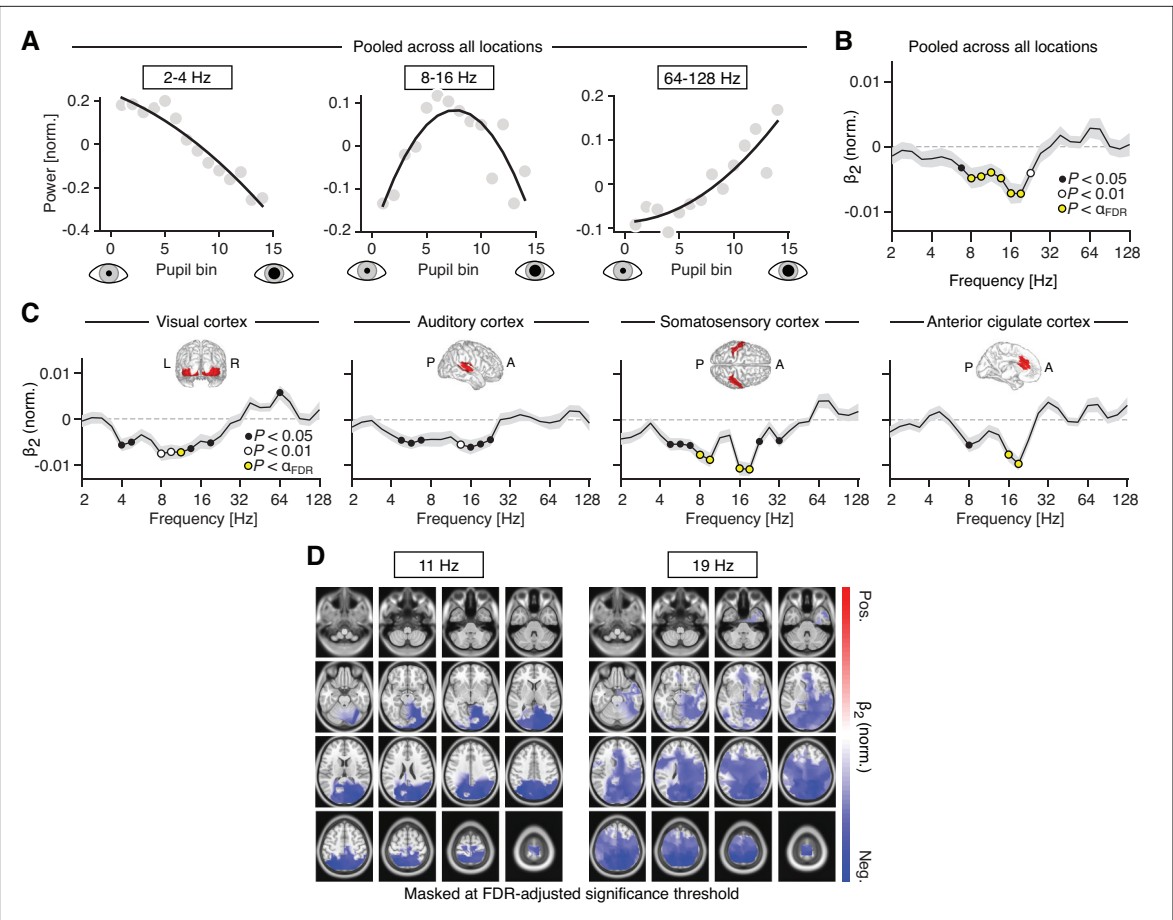

**Figure 5.** Nonlinear relations between pupil-linked arousal and band-limited power fluctuations. (**A**) Normalized spectral power, averaged across three frequency ranges (2–4, 8–16, and 64–128 Hz; from left to right), as a function of mean pupil diameter. (**B**) Normalized coefficient of the quadratic term ($\beta_2$; see Materials and methods for details), averaged across all subjects and atlas regions. Black dots denote p < 0.05, white dots denote p < 0.01 and yellow dots denote p < 0.005 (FDR-adjusted significance threshold with q = 0.1). The gray shaded area depicts the standard error of the mean across subjects. Positive and negative values are indicative of a U-shaped and inverted U-shaped relationship, respectively. (**C**) Normalized $\beta_2$ coefficient, averaged across all subjects, separately for four regions of interest: visual cortex, auditory cortex, somatosensory cortex, and anterior cingulate cortex (from left to right). Insets show the approximate extent of the regions of interest. Yellow dots denote p < 0.004 (FDR-adjusted significance threshold with q = 0.1). (**D**) Spatial distribution of the coefficient of the quadratic term (see Materials and methods for details), at center frequencies 11 Hz (left) and 19 Hz (right). Masked at the FDR-adjusted significance threshold with q = 0.1 (11 Hz: α = 0.003; 19 Hz: α = 0.0081). All p-values were obtained from two-tailed t-tests.

The online version of this article includes the following figure supplement(s) for figure 5:

**Figure supplement 1.** Nonlinear relations between pupil-derivative and band-limited power fluctuations.

lobule (*Figure 5D*, center left). The same relationship mapped at a higher center frequency (19 Hz) comprised additional bilateral parietal and temporal regions (*Figure 5D*, right).

We also found some evidence for quadratic relationships when analyzing the pupil derivative instead of the pupil diameter, however exhibiting different spatial patterns and a less pronounced inverted U-shape for the intermediate frequency range (*Figure 5—figure supplement 1*).

Taken together, we found evidence for a inverted U relationship between pupil-linked arousal and band-limited cortical activity that was confined to the classical alpha/beta frequency band (8–32 Hz) and parieto-occipital cortices. Put differently, cortical activity in this frequency range, and in the respective cortical regions, was maximal at medium pupil diameters that indicate intermediate optimal levels of arousal. Activity in surrounding low- and high-frequency ranges showed exclusively linear effects.

## Pupil-linked arousal also predicts changes in aperiodic components of cortical population activity

So far, we have shown how pupil-linked arousal covaries with band-limited cortical activity. However, cortical electrophysiological processes can also be characterized by a broadband, or aperiodic component. This component can be understood as the decay of power with increasing frequency and is therefore also known as the 1/f component. The slope of the 1/f component reflects fluctuations in attentional state (*Waschke et al., 2021*) and has been suggested to track the ratio between excitation and inhibition (in short: E/I) in the underlying neuronal circuits (*Gao et al., 2017*). As neuromodulators linked with the regulation of arousal have been shown to change cortical E/I (*Martins and Froemke, 2015*; *Pfeffer et al., 2018*; *Pfeffer et al., 2021*), we next investigated the relation between intrinsic fluctuations in pupil diameter (as well as its temporal derivative) and the aperiodic component in the power spectrum. Our goal was to characterize correlations between intrinsic fluctuations in pupil-linked arousal and the slope of the aperiodic component, as a potential marker of cortical E/I.

To this end, we parameterized the power spectra of consecutive, overlapping data bins of 2 s length of the source-projected MEG data in a frequency range from 3 to 40 Hz, separately for all source locations (*Donoghue et al., 2020*). Simulations of a biologically plausible neural network (*Trakoshis et al., 2020*) as well as empirical insights from optogenetic stimulations in neonatal mice (*Chini et al., 2021*) show that spectral slopes extracted from the chosen frequency range are in fact

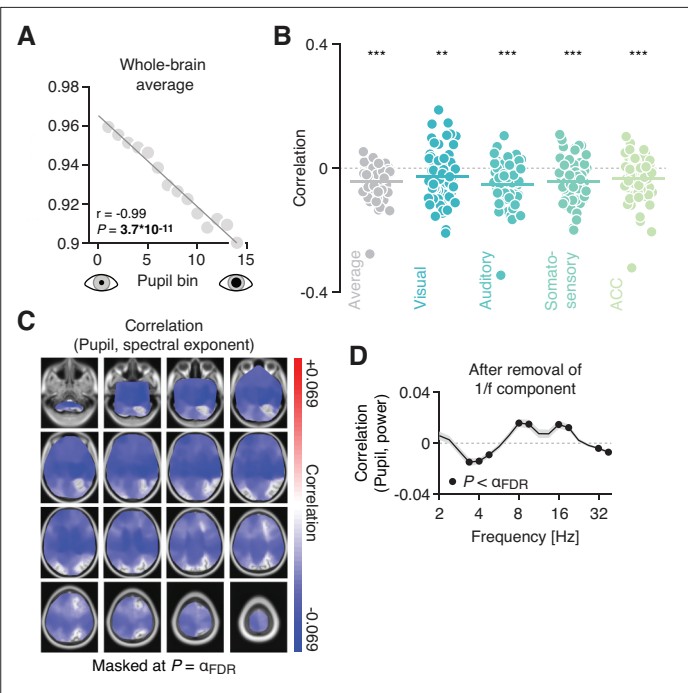

**Figure 6.** Pupil-linked arousal correlates with slope of the aperiodic component of the power spectrum. (**A**) Spectral exponent (i.e., the slope of the aperiodic component) for 14 pupil bins (sorted from small to large pupil diameter). (**B**) Coefficients of the correlation between the spectral exponent and pupil diameter fluctuations, averaged across space (black dots, showing the individual subjects) and for the four regions of interest: visual cortex, auditory cortex, somatosensory cortex, and anterior cingulate cortex. The horizontal lines highlight the mean correlation coefficient (***p < 0.001, **p < 0.01; two-tailed paired t-tests). (**C**) Spatial map of the correlation between the slope of the aperiodic component and intrinsic fluctuations in pupil diameter. Spatial maps are masked at the FDR-adjusted significance threshold of $\alpha$ = 0.009 (with q = 0.1); p-values were obtained from a two-tailed paired t-test. (**D**) Correlations between fluctuations in pupil diameter and band-limited power, after removing the aperiodic component. The frequency axis is truncated, due to fitting the slope of the power spectrum only in the 3–40 Hz range. Black dots denote significant values (filled circles p < 0.012; FDR-adjusted significance threshold with q = 0.1; two-tailed paired t-test).

The online version of this article includes the following figure supplement(s) for figure 6:

**Figure supplement 1.** Pupil derivative correlates with aperiodic component of the power spectrum.

sensitive to changes in the underlying ratio between excitation and inhibition. The resulting parameter estimates described the periodic and the aperiodic components per epoch (data segmentation as in the previous section; see Materials and methods). Again, for this analysis we shifted the pupil time series forward by 930 ms relative to the MEG signal (no shift was applied in the corresponding pupil derivative analysis). The analysis yielded the parameterized power spectrum (for each source region) and average pupil diameter values (or pupil derivative) for each of the temporal bins.

First, we sorted all bins based on mean pupil diameter (small to large) and plotted the spectral exponent, a measure that expresses the steepness of the slope of the aperiodic component of the power spectrum, against it. The spectral exponent decreased strongly linearly as a function of pupil diameter (*Figure 6A*), indicative of a shallower decay of power as a function of frequency (see Materials and methods for details). We next correlated the fluctuations in the estimated spectral exponent with fluctuations in pupil diameter across all bins (i.e., without prior sorting and averaging within bins). This resulted in a significant negative correlation when averaged across all source locations (*Figure 6B*, gray dots), with little variability when zooming in on four regions of interest (visual cortex: mean $r = -0.0278$; $p = 0.0014$; auditory cortex: mean $r = -0.0518$, $p = 4.5*10^{-10}$; somatosensory cortex: mean $r = -0.0421$, $p = 1.09*10^{-5}$; anterior cingulate cortex: $r = -0.0336$, $p = 1.09*10^{-5}$; two-tailed paired t-tests against zero). In fact, the negative correlation between the spectral exponent and mean pupil diameter for the corresponding segment was significant in the vast majority of brain regions (8137/8799 of source locations, or 92.5%, after adjusting for a false discovery rate (FDR) of 0.1; *Figure 6C*). A correlation was notably absent in several extrastriate occipital cortices.

We found a similarly widespread negative correlation between the spectral exponent and the pupil derivative. Although weaker in magnitude, this effect additionally encompassed the entire occipital cortex and showed stronger local maxima in parietal cortices, when compared with the correlation with pupil (*Figure 6—figure supplement 1A-C*).

In sum, these results speak to a global effect of pupil-linked arousal on the excitation-inhibition ratio across the brain, with few potential exceptions (occipital cortices). Minor differences between pupil- and pupil-derivative-based spatial patterns of these effects may again point toward the involvement of different neuromodulators.

## Couplings between pupil-linked arousal and periodic components of cortical population activity are not induced by couplings with aperiodic components

Given the pupil-linked changes in the slope of the (aperiodic component of) the cortical power spectrum, one concern is that these introduced spurious changes in band-limited power (*Donoghue et al., 2020*), thus potentially producing spurious correlations between fluctuations in pupil diameter and band-limited power. To test whether the correlations between pupil and spectral power reported above were merely a result of changes in the slope of the power spectra, we subtracted the fitted aperiodic component from the empirical power spectra (in the range from 3 to 40 Hz), separately for each time segment and source location. Subsequently, we correlated the residual power estimates with pupil diameter, across all segments.

The pattern of correlations between pupil diameter and power remained largely unchanged (*Figure 6D*): Fluctuations in pupil diameter were negatively correlated with activity in the low frequencies, around 4 Hz, as well as with activity in a higher frequency range around 32 Hz. Band-limited activity in the range between 8 and 16 Hz exhibited robust positive correlations. Correlations between residual power and pupil derivative, on the other hand, were strongly reduced in magnitude and not significant for most frequencies (*Figure 6—figure supplement 1D*).

These control analyses confirm that the couplings between pupil (and pupil-derivative) and band-limited cortical activity reported in the preceding sections were genuine reflections of arousal-related modulations of periodic cortical activity with distinct effects for different frequency bands.

## Discussion

Pupil-linked arousal exerts pivotal influences on the states of sensory cortices (*McGinley et al., 2015a*; *Reimer et al., 2014*; *Vinck et al., 2015*) and associative cortical regions (*Joshi and Gold, 2020*). However, subcortical arousal systems also project to a wealth of other regions across the cerebral

cortex. This warranted a detailed investigation of how these projections shape cortical population activity. To achieve this, we exhaustively mapped the relationship between intrinsic fluctuations in pupil-linked arousal and resting-state human MEG-recorded cortical activity. In a large sample (*N* = 81), our mapping revealed diverse and consistent arousal effects on band-limited, as well as aperiodic cortical activity.

## Arousal modulates cortical activity across space, time, and frequencies

Our findings are consistent with previously reported links between both locomotion-related and pupil-related changes in arousal and LFP recordings in sensory cortices of rodents (*McGinley et al., 2015a*; *Reimer et al., 2014*; *Reimer et al., 2016*). Briefly put, periods of elevated arousal were accompanied by a decrease in low- and an increase in high-frequency activity, similar to cortical state changes typically observed during increased attention (*Harris and Thiele, 2011*; *Schwalm and Rosales Jubal, 2017*). The observed temporal relationships also align with the lags between manipulations of arousal and pupil dilations of around 500–1000 ms in humans (*Hoeks and Levelt, 1993*; *Wierda et al., 2012*) and an ~500 ms delay between LC activation and peak pupil dilation in non-human primates (*Joshi et al., 2016*). In rodents, *Reimer et al., 2016*, and *Breton-Provencher and Sur, 2019*, reported peak correlations between noradrenergic axonal activity and pupil diameter at around 900 ms, whereas the correlation between BF-regulated cholinergic activity and pupil diameter peaked around 500 ms (*Reimer et al., 2016*).

Here, we report comparable peak lags for low- (832 ms) and high-frequency activity (562 ms), suggesting that different neuromodulatory systems may drive correlations in different frequency ranges in the human brain. Future research using brain-wide maps of the distribution of neuromodulatory receptors (*van den Brink et al., 2019*) may help to further disentangle the differential contributions of NE and ACh as well as their various receptor subtypes. Additionally, this may elucidate the roles of serotonergic and dopaminergic neuromodulation, which also influence pupil size (*Cazettes et al., 2021*; *de Gee et al., 2017*). Further, we note that our analyses and interpretations focus on arousal-related neuromodulatory influences on cortical activity, whereas recent work also supports a reverse 'top-down' route, at least for frontal cortex high-frequency activity on LC spiking activity (*Totah et al., 2021*).

The low-frequency regime referred to in rodent work (2–10 Hz; e.g., *McGinley et al., 2015b*) includes activity that shares characteristics with human alpha rhythms (3–6 Hz; e.g., *Nestvogel and McCormick, 2022*; *Senzai et al., 2019*). The human equivalent however clearly separates from activity in lower frequency bands and, here, showed idiosyncratic relationships with pupil-linked arousal. Specifically, a negative correlation, localized to somatomotor cortices and specific to alpha-band (8–12 Hz) activity, peaked earlier than a positive correlation that localized to occipital visual cortices and encompassed alpha and beta bands (see *Figures 3C and 4D*). Of all effects found, the effect in visual cortices had the shortest lag with respect to pupil dilation (~200 ms).

This short-lag positive correlation may stem from interactions between the LC and regions of the thalamus (*McCormick, 1989*; *McCormick et al., 1991*; *Stitt et al., 2018*), in particular those implicated in the generation of occipital alpha-band activity via thalamo-cortical feedback loops (e.g., the pulvinar; *Saalmann et al., 2012*). These loops may therefore be sensitive to noradrenergic (*Dahl et al., 2022*) or even BF regulated cholinergic modulation (*Nestvogel and McCormick, 2022*; *Zagha and McCormick, 2014*). Both scenarios posit indirect multi-synaptic routes to the neuromodulation of alpha activity that may delay the lag relative to other effects and put it closer to the time of maximum pupil dilation. Another, not mutually exclusive, possibility is that the arousal-dependent low-lag alpha-/beta-band activity is linked to the activity of a distinct subpopulation of GABAergic neurons in the LC, which have been shown to exhibit a temporally closer relationship with pupil diameter compared to the noradrenergic neurons (*Breton-Provencher and Sur, 2019*).

Seemingly contradicting the present findings, previous work on task-related EEG and MEG dynamics reported a negative relationship between pupil-linked arousal and alpha-range activity in occipito-parietal sensors during visual processing (*Meindertsma et al., 2017*) and fear conditioning (*Dahl et al., 2020*). Note however that results from task-related experiments, that focus on evoked changes in pupil diameter rather than fluctuations in tonic pupil size, cannot be directly compared with our findings. Similar to noradrenergic neurons in LC (*Aston-Jones and Cohen, 2005*), phasic pupil responses exhibit an inverse relationship with tonic pupil size (*Knapen et al., 2016*). This means

that on trials with larger baseline pupil diameter (e.g., during a pre-stimulus period), the evoked (phasic) pupil response will be smaller and vice versa. Therefore, a negative correlation between alpha-band activity in the visual cortex and task-related phasic pupil responses does not preclude a positive correlation with tonic pupil size during baseline or rest, as reported here. In line with this, *Whitmarsh et al., 2021*, found a negative relationship between alpha-activity and pupil size in the somatosensory cortex that agrees with our finding. Although using an event-related design to study attention to tactile stimuli, this relationship occurred in the baseline, that is, before observing any task-related phasic effects on pupil-linked arousal or cortical activity.

Note that reported changes in alpha activity in the visual cortex cannot be trivially explained by pupil dilation altering retinal illumination. In that case, one would expect a negative relationship, that is, decreasing alpha power with increasing pupil size (e.g., *Donner and Siegel, 2011*). Moreover, changes in alpha activity preceded changes in pupil dilation, albeit with a shorter lag compared to activity in other frequency ranges (see *Figure 3*).

Correlations between cortical activity and the pupil derivative largely mirrored these results, although most links were found to be weaker in magnitude. This is likely due to the fact that both signals (pupil diameter as well as its temporal derivative) correlate to some extent with noradrenergic and cholinergic activity (*Reimer et al., 2016*).

## An arousal-triggered cascade of activity in the resting human brain

The temporal profile of the diverse relationships between pupil-linked arousal and local cortical activity described above may point toward a cascade of interrelated effects that the resting brain undergoes routinely. Several scenarios could produce such a cascade: First, episodes of arousal could arise seemingly at random, from underlying complex nonlinear neural dynamics (*Kringelbach and Deco, 2020*; *Robinson et al., 2003*). Second, they could be coupled to other physiological processes, such as respiration (*Kluger and Gross, 2021*; *Zelano et al., 2016*). Respiration has an average base rate of ~0.2 Hz (*Del Negro et al., 2018*; *Fleming et al., 2011*), similar to pupil fluctuations (*Bouma and Baghuis, 1971*; *Pomè et al., 2020*; *Turnbull et al., 2017*). As a third scenario, the observed cascade may imply a specific cognitive sequence: Most participants will likely have 'mind-wandered' or engaged in 'task-unrelated thoughts' (*Groot et al., 2021*; *Karapanagiotidis et al., 2020*) while resting. The sequence of effects we observe may therefore be a transient re-orientation toward the external sensory environment – a 'monitoring sweep'. This is consistent with the increase in pupil-linked arousal, and its inverse relationship with an early dip in low-frequency activity, that is, the cessation of a globally synchronized cortical state, and agrees with a concurrent widespread increase in high-frequency cortical activity that signifies increased cortical excitability to process external sensory input (*Hanslmayr et al., 2011*; *Harris and Thiele, 2011*).

One aspect of mind-wandering is the retrieval of information from memory that heavily relies on theta rhythms in the hippocampus (*Siegle and Wilson, 2014*). Here, we localized the strongest pupil (and pupil-derivative) correlations with theta-range power fluctuations in bilateral hippocampal regions (see *Figure 4D*), suggesting an arousal modulation of memory processes. In support, hippocampal LC-NE modulation has long been established in the rodent brain (*Sara, 2009*; *Segal and Bloom, 1976a*; *Segal and Bloom, 1976b*), and *McGinley et al., 2015a*, reported negative correlations between pupil-linked arousal and the occurrence of theta-range hippocampal ripple events. Moreover, a recent human neuroimaging study found that pupil-linked arousal modulated blood oxygenation in (para-) hippocampal regions during memory processes (*Clewett et al., 2018*).

Further on, the observed sequence of effects concludes with a peak in alpha/beta activity that is triggered roughly at the time of maximum correlations of arousal with low- and high-frequency activity. The increase in synchronized alpha/beta activity marks the return to a strong regime where external sensory (visual) input is attenuated, potentially indicating a shift to an introspective focus. A strong alpha rhythm thereby imposes widespread bouts of 'pulsed inhibition' on visual cortex that suppress local gamma band activity (*Jensen and Mazaheri, 2010*; *Haegens et al., 2011*; for a review, see *Van Diepen et al., 2019*). This relationship between alpha and gamma activity has been well described in non-human primate (*Spaak et al., 2012*) and human visual cortices, with a role for cholinergic neuromodulation (*Bauer et al., 2012*). Corroborating this scenario, we also observe a decrease in local gamma-band activity in visual cortices.

The three scenarios described here are not mutually exclusive and may explain one and the same phenomenon from different perspectives. Further, it remains possible that the sequence we observe comprises independent effects with specific timings. A pivotal manipulation to test these assumptions will be to contrast the observed sequence with other potential coupling patterns between pupil-linked arousal and cortical activity during different behavioral states. In fact, *Kucyi and Parvizi, 2020*, found spontaneous peaks of high-frequency band activity (>70 Hz) in the insular cortex of three resting surgically implanted patients that preceded pupil dilation by ~500 ms – a time range that is consistent with the lag of our cross-correlation between pupil size and high-frequency (>64 Hz) activity (see *Figure 3B*). Importantly, they showed that this sequence mimicked a similar but more pronounced pattern during task performance. Given the purported role of the insula (*Menon and Uddin, 2010*), this finding lends support to the idea that spontaneous covariations of pupil size and cortical activity signal arousal events related to intermittent 'monitoring sweeps' for behaviorally relevant information.

## Inverted U relationship between periodic cortical activity and pupil-linked arousal

Nonlinear relations between arousal and other factors are well characterized, most prominently in the form of the Yerkes-Dodson law, which posits an inverted U-shaped relation between arousal and cognitive performance (*Yerkes and Dodson, 1908*). More recently, an inverted U relation has also been identified for pupil-linked arousal fluctuations and detection performance and low-frequency (2–10 Hz) membrane potential fluctuations in primary auditory cortex (*McGinley et al., 2015a*). Here, we found a pronounced and comparable inverted U-shaped relationship between power in the alpha and beta bands, and pupil and pupil derivative alike: The inverted U alpha-band nonlinearity was most strongly expressed in inferior parietal cortex, extending toward higher visual as well as temporal regions in the beta band. In contrast to *McGinley et al., 2015a*, we found no evidence for an U-shaped relationship between pupil and high-frequency (50–100 Hz) activity.

The inverted U-shaped relationship between arousal and behavior may result from the differential activation of neuromodulatory receptors in particular noradrenergic adrenoreceptors with varying affinity (*Berridge and Spencer, 2016*) or nicotinic and muscarinic ACh receptors (*Bentley et al., 2011*). Similar mechanisms may therefore underlie the quadratic relationship observed here. More specifically, in the case of NE, the activation of high-affinity $\alpha_2$ receptors at medium arousal level may (directly or indirectly) lead to increased alpha- and beta-band activity. At elevated arousal levels, on the other hand, correspondingly high levels of NE lead to the activation of low-affinity $\alpha_1$ receptors, possibly resulting in a decrease of alpha- and beta-band activity (*Buzsáki et al., 1991*).

From a functional perspective, an inverted U relationship seems paradoxical. One would expect alpha activity to approach a minimum at intermediate states of pupil-linked arousal, that is, show a U-shaped relationship instead, to optimally facilitate the processing of external sensory input (*Jensen and Mazaheri, 2010*). However, our participants were resting, while neither receiving nor expecting visual input. In this situation an inward focus of attention suppressing external visual information may have reversed the expected relationship (also see *Hong et al., 2014*).

Of note, we find evidence for both, monotonic positive and inverted U shape relationships between alpha/beta activity and pupil-linked arousal in spatially coinciding cortical sources (cp. *Figures 4D and 5C*). Earlier research has suggested that parieto-occipital cortices harbor at least two distinct alpha rhythms (*Barzegaran et al., 2017*; *Keitel and Gross, 2016*) with separable characteristics (*Benwell et al., 2019*) that may serve different functions (*Capilla et al., 2014*; *Sokoliuk et al., 2019*). Therefore, spatially overlapping monotonic and inverted U effects may reflect distinct influences of arousal on the generative processes underlying these separate alpha rhythms.

## Arousal modulation of cortical excitation-inhibition ratio

In addition to interactions between pupil-linked arousal and band-limited activity, we show that pupil diameter correlates systematically with aperiodic brain activity. Recent theoretical and empirical evidence suggests that the slope of the aperiodic component of neural power spectra, the spectral exponent, reflects the ratio between excitatory and inhibitory neuronal processes in the underlying neuronal circuits (*Colombo et al., 2019*; *Gao et al., 2017*).

Here, we report a negative monotonic relation between pupil diameter and the spectral exponent. In other words, periods of elevated arousal were accompanied by a flattening of the power spectrum.

In simulated neural networks, a shallower slope of the aperiodic component is related to increased excitation relative to inhibition (*Gao et al., 2017*; *Trakoshis et al., 2020*). Thus, increased arousal (i.e., dilated pupils) may co-occur with increased E/I in distributed neuronal circuits, whereas periods of low arousal (i.e., constricted pupils) would indicate states of relatively decreased E/I. This interpretation is also consistent with previous findings arguing that NE increases cortical E/I (*Pfeffer et al., 2018*; *Pfeffer et al., 2021*), possibly through a decrease in inhibition (*Froemke, 2015*; *Martins and Froemke, 2015*). We found this relationship of E/I and pupil-linked arousal in most cortical regions, thus suggesting a global and uniform effect that only excluded some occipital visual cortices (see *Figure 6C*).

The absence of this effect in visual cortices may explain why *Kosciessa et al., 2021*, found no relationship between pupil-linked arousal and spectral slope when investigating phasic pupil dilation in response to a visual stimulus during task performance. However, this behavioral context, associated with different arousal levels, likely also changes E/I in the visual cortex when compared with the resting state (*Pfeffer et al., 2018*).

In sum, pupil-linked arousal may provide a window into yet another cortical circuit property, cortical E/I, relevant for cognitive computation (*Cavanagh et al., 2020*; *Kosciessa et al., 2021*; *Lam et al., 2017*; *Murphy and Miller, 2003*; *Pettine et al., 2021*; *Pfeffer et al., 2021*).

## Conclusion

We exhaustively mapped how fluctuations in pupil-linked arousal and cortical activity covary in the human brain at rest. The diverse relationships we describe in our data suggest profound influences of arousal on cortical states and challenge a global and unspecific role for arousal neuromodulation in cortical function. Instead, our results support the view of specific neuromodulatory influences that differ depending on the targeted cortical region, can express in varying frequencies of band-limited activity or as broadband effects, and vary in their timing. Further, they largely agree with similar findings of a recent independent report (*Podvalny et al., 2021*). The present data provide the basis for studying how these influences change under cognitive task demands and when engaging in different behaviors. In fact, neuromodulatory effects are often neglected when studying the cortical population activity underlying cognitive function – given a central role for arousal, our results argue for including it as a covariate in future studies.

## Materials and methods

### Participants and data acquisition (Hamburg)

Thirty healthy human participants (16 women, mean age 26.7, range 20–36) participated in the study after informed consent. All included participants were healthy, with no current or previous diagnosis of psychiatric or neurological disorder (full list of exclusion criteria can be found in *Pfeffer et al., 2021*). The study was approved by the Ethics Committee of the Medical Association Hamburg (approval number PV4648). Two participants were excluded from analyses, one due to excessive MEG artifacts, the other due to not completing all recording sessions (see below). Thus, we report results from *N* = 28 participants of which 15 were women. The present dataset is a subset of a larger dataset that entailed selective pharmacological manipulations (*Pfeffer et al., 2018*; *Pfeffer et al., 2021*). For the present article, only data from the placebo/resting-state condition was analyzed.

Each participant completed three experimental sessions (scheduled at least 2 weeks apart), consisting of drug or placebo intake at two time points, a waiting period of 3 hr, and an MEG recording session. During the recordings, participants were seated on a chair inside a magnetically shielded chamber. Each recording session consisted of two resting-state measurements as well as four blocks consisting of two variants of a behavioral task. Each block was 10 min long and followed by a short break of variable duration.

Brain activity was recorded using a whole-head CTF 275 MEG system (CTF Systems, Inc, Canada) at a sampling rate of 1200 Hz. Eye movements and pupil diameter were recorded with an MEG-compatible EyeLink 1000 Long Range Mount system (SR Research, Osgoode, ON, Canada) and electrocardiogram as well as vertical, horizontal, and radial EOG was acquired using Ag/AgCl electrodes. During all recordings, the participants were instructed to keep their eyes open and fixate a green

fixation dot in the center of a gray background of constant luminance, which was projected onto a screen from outside the magnetically shielded recording chamber at a refresh rate of 60 Hz.

MRI anatomical scans were acquired in supine position on a Magnetom Trio (Siemens). MEG and MRI coordinate systems were co-registered based on anatomical landmarks (preauricular points, nasion).

## Participants and data acquisition (Münster)

Forty right-handed volunteers (21 women, age 25.1 ± 2.7 year (mean ± SD), range 21–32 years) participated in the study. All participants reported having no respiratory or neurological disease and gave written informed consent prior to all experimental procedures. The study was approved by the local ethics committee of the University of Münster (approval number 2018–068f-S).

Participants were seated upright in a magnetically shielded room while we simultaneously recorded 5 min of MEG and eye tracking data. MEG data was acquired using a 275 channel whole-head system (OMEGA 275, VSM Medtech Ltd., Vancouver, Canada) at a sampling frequency of 600 Hz. During MEG recordings, head position was continuously tracked online by the CTF acquisition system. To this end, a set of three coils were placed on the head of the participant. Pupil area of the right eye was recorded at a sampling rate of 1000 Hz using an EyeLink 1000 plus eye tracker (SR Research). During recording, participants were to keep their eyes on a fixation cross centered on a projector screen placed in front of them. To minimize head movement, participants' heads were stabilized with cotton pads inside the MEG helmet.

MRI acquisition of anatomical data for source reconstructions (Siemens 3T Prisma with a 20-channel head coil) was conducted in supine position to reduce head movements and gadolinium markers were placed at the nasion as well as left and right distal outer ear canal positions for landmark-based co-registration of MEG and MRI coordinate systems.

## Participants and data acquisition (Glasgow)

MEG resting-state data were acquired for 24 healthy, right-handed participants (9 women; age 23.5 ± 5.5 years (mean ± SD), range 18–39 years). The study was approved by the local ethics committee (University of Glasgow, College of Science and Engineering; approval number 300140078). Participants gave written informed consent prior to testing.

Recordings were obtained with a 248-magnetometer whole-head MEG system (MAGNES 3600 WH, 4-D Neuroimaging), set to a sampling rate of 1017 Hz, and an EyeLink 1000 eye tracker (SR Research), sampling the pupil area of the left eye at 1000 Hz, both situated in a magnetically shielded room (Vacuumschmelze). Head position was measured at the start and end of the recording. To this end, a set of five coils were placed on the head of the participant. Coil positions and head shape were digitized using a FASTRAK stylus (Polhemus Inc, VT). During recordings, participants sat upright and fixated on a green point (measure) projected centrally on a screen with a DLP projector (60 Hz refresh rate) for a minimum of 7 min.

Resting-state recordings were part of a study into audio-visual speech processing that was recorded in two sessions. Audio-visual data have been published elsewhere (*Keitel et al., 2018*; *Keitel et al., 2020*). In this study, the resting-state block was recorded as the last block of the second session after seven to nine experimental blocks, or more than 65 min into the session. Note that the resting-state data have not been reported in any of the previous publications.

Data of two participants were excluded from analysis after screening recordings, one due to strong intermittent environmental artifacts in their MEG recordings and another due to a technical fault in tracking the pupil.

MRI anatomical scans were acquired on a 3T Siemens Tim Trio system with a 12-channel head coil in supine position. MEG and MRI coordinate systems were co-registered using the Polhemus-tracked headshape.

## MEG preprocessing (Hamburg)

The sensor-level MEG signal was cleaned of extra-cranial artifacts using semiautomatic artifact detection routines implemented in the FieldTrip toolbox (*Oostenveld et al., 2011*). Artifactual samples were identified through visual inspection for each channel and were marked and removed across all channels (±500 ms). Subsequently, the data were downsampled to 400 Hz split into low ([0.5–2]–40 Hz;

the lower cutoff was variable across (but identical within) subjects at 0.5, 1, or 2 Hz) and high (>40 Hz) frequency components, using a fourth-order Butterworth filter. The separate data segments were independently submitted to independent component analysis (ICA; *Hyvärinen, 1999*). The split into low- and high-frequency data aimed to facilitate the detection of sustained muscle artifacts especially present in the high-frequency range. Artifactual-independent components were identified through visual inspection of their topography and power spectra. On average, 23 ± 14 components (mean ± SD) were subtracted from the raw data (placebo condition only). After artifactual components were removed, the low- and high-frequency segments were combined into one single dataset.

## MEG preprocessing (Münster)

MEG preprocessing was done in Fieldtrip (*Oostenveld et al., 2011*) for MATLAB (The MathWorks Inc). First, noisy recording channels were identified by visual inspection and rejected from further analyses. Next, we adapted the synthetic gradiometer order to the third order for better MEG noise balancing (ft_denoise_synthetic). Power line artifacts were removed using a discrete Fourier transform (DFT) filter on the line frequency of 50 Hz and all its harmonics (including spectrum interpolation; *dftfilter* argument in *ft_preprocessing*). Next, we applied ICA (*Hyvärinen, 1999*) on the filtered data to capture eye blinks and cardiac artifacts (*ft_componentanalysis* with 32 extracted components). On average, artifacts were identified in 2.35 ± 0.83 components (M ± SD) per participant and removed from the data. Finally, cleaned MEG data were downsampled to 400 Hz.

## MEG preprocessing (Glasgow)

For MEG and eye tracking data preprocessing and analysis, we used MATLAB (The MathWorks Inc), involving in-house MATLAB routines and the FieldTrip toolbox (*Oostenveld et al., 2011*). The MEG signal was resampled to the sampling rate of the eye tracker (1000 Hz) and denoised using in-built MEG reference sensors. We rejected between 2–27 noisy MEG channels (median = 7.5 ± 4 absolute deviation) by visual inspection using FieldTrip's *ft_rejectvisual*. Continuous MEG recordings were then screened for noisy segments. High-pass filtered (2 Hz cutoff, fourth-order Butterworth, forward-reverse two-pass) MEG time series were subjected to an ICA (*Hyvärinen, 1999*). Prior to the ICA, a principal component analysis (PCA) projected MEG data into a 32-dimensional subspace. For each participant, we visually identified two to four components (median = 3) capturing eye (blinks, movements) and heart-beat artifacts. Finally, resampled continuous MEG recordings were projected through the subspace spanned by the remaining components.

MEG and eye tracker time axes were realigned by cross-correlating traces of horizontal and vertical eye movements that were simultaneously recorded by the eye tracker and the MEG. The lag corresponding to the maximum of the cross-correlation functions of horizontal and vertical eye movement traces was used as offset for the realignment.

## Pupil preprocessing (all sites)

Pupil area traces were converted to pupil diameter to linearize our measure of pupil size. Using additional MATLAB routines (available from https://github.com/anne-urai/pupil_preprocessing_tutorial, as used in *Urai et al., 2017*), blinks were identified by an automatic (and visually validated) procedure and linearly interpolated. The number of blinks and, therefore, the percentage of the interpolated data points varied across MEG sites: in total, 17.9% of the Glasgow pupil time series (standard deviation: 17.9%; range: 1.3–39.8%, with one outlier of 83.3%), 11.9% of the Hamburg pupil time series (standard deviation: range; 11.3%; 0–37.1%) and 22.9% of the Münster pupil time series (standard deviation: 14.9%; range: 0–56.7%) were interpolated. In a second pass, a similar procedure was used on smaller eye tracker artifacts (spikes, jumps). Next, canonical responses to blinks and saccades were estimated and then removed from pupil time series (*Hoeks and Levelt, 1993*; *Knapen et al., 2016*; *Wierda et al., 2012*). To that end, pupil time series were band-pass filtered (pass band: 0.005–2 Hz, second-order Butterworth, forward-reverse two-pass), then downsampled to a sampling rate of 400 Hz. The low-pass cutoff was set to 2 Hz to coincide with the lowest frequency of interest regarding power fluctuations in cortical oscillations in our analysis. It captured the typical time scales of arousal-linked pupillary phenomena such as the Hippus (around 0.2 Hz) and the canonical response to arousing events (<1 Hz given an average $t_{max}$ = 0.93 s for peak dilation). Further, data of *McGinley et al., 2015a*, suggested that neuromodulation has the most profound influences on pupillary dynamics in

the range below <1 Hz. Finally, we computed the first-order derivative of each band-pass filtered and downsampled pupil trace.

## Spectral decomposition of pupil time series (all sites)

We conducted additional spectral analyses of pupil time series for the purpose of cross-laboratory comparisons only – see *Figure 1—figure supplement 2*. All analyses reported in the Results are based on the pupil diameter time series (or their derivative).

The spectral content of pupil traces was evaluated using the multi-taper approach as implemented in FieldTrip. We used the default settings, except for a spectral smoothing of 0.035 Hz, and zero-padding time series to $2^{19}$ (minimum power of 2 that exceeded the length of any pupil trace) to unify frequency resolutions across centers and traces of different lengths. Power spectra were calculated on individually standardized (z-scored) pupil traces for the 0.005–2 Hz range, sampled in logarithmic steps.

Resulting power spectra were also subjected to the spectral parameterization toolbox ('FOOOF'; *Donoghue et al., 2020*) to extract Hippus peak frequency as well as the exponent describing the spectral slope as a means to characterize the aperiodic activity. We used the FOOOF MATLAB wrapper with the default settings, except setting the maximum number of peaks ('max_n_peaks') to 3, the peak detection threshold ('peak_threshold') to 0.5, and using a peak width range of [0.1, 0.5] Hz ('peak_width_limits').

## MEG spectral analysis (all sites)

In order to derive spectral estimates from the sensor-level data, we followed an approach outlined previously (*Hipp et al., 2012*), based on Morlet's wavelets (*Tallon-Baudry and Bertrand, 1999*):

$$w\left(t,f\right) = \sigma_t \sqrt{\pi}^{-1/2} e^{\frac{-t^2}{2\sigma_t^2}} e^{-i2\pi ft} \tag{1}$$

Following *Hipp et al., 2012*, we derived spectral estimates for 25 logarithmically spaced center frequencies, covering a broad frequency range from 2 to 128 Hz. Each frequency band was defined as the halve-octave band around the respective center frequencies, thus adapting spectral and temporal widths of wavelets to accommodate wider bands and finer temporal resolution for higher frequencies. Spectral estimates were derived for successive temporal windows with an overlap of 80%. Windows that contained periods marked as artifactual (see Preprocessing) were discarded from all further analyses. From the spectral estimates, sensor-level power envelopes were obtained through:

$$Y\left(t,f\right) = \left| X_{sens}\left(t,f\right) \right|^2 \tag{2}$$

where $X_{sens}$ denotes the complex spectral estimates for segment $t$ and frequency $f$.

## Microsaccade-related changes in cortical activity and pupil size (all sites)

Microsaccades were detected from the horizontal and vertical eye movement coordinates using the microsaccade detection algorithm described in *Engbert and Kliegl, 2003*. The minimum saccade duration and the threshold velocity were set to five samples (equal to 12.5 ms) and six, respectively. The chosen parameter values correspond to the default values used in the original paper. Estimates of spectral power were obtained from a short-time Fourier transform (using MATLAB's 'pwelch' function, with each segment tapered by a Hann window) with sliding windows of 0.5 s length and a temporal shift of 50 ms. Sensor-level MEG data were analyzed from 0.25 s prior (effectively from –0.5 to 0 s) to 1.75 s following (effectively from 1.5 to 2 s) the onset of a microsaccade. Microsaccade-related changes in spectral power were defined as the percentual change in power relative to the pre-microsaccade baseline interval, defined as the time bin centered at $t = –0.25$ s. For each time bin, pupil size values were tapered with the same Hann window and the resulting values were summed. Percentual change in pupil size were then computed equivalently to the changes in spectral power.

## MEG source reconstruction (all sites)

Accurate source models were generated using the FieldTrip interface to SPM8 (*Litvak et al., 2011*) and the Freesurfer toolbox (*Fischl, 2012*). Source models were based on individual T1-weighted structural MRIs. Anatomical scans were co-registered with the MEG coordinate system using a semiautomatic procedure (*Gross et al., 2013*), then segmented and linearly normalized to a template brain (MNI space). We used a single shell as the volume conduction model (*Nolte, 2003*).

We projected sensor-level data into source space using frequency-specific DICS (Dynamic Imaging of Coherent Sources) beamformers (*Gross et al., 2001*) with a regularization parameter of 5% and optimal dipole orientation (singular value decomposition method). As the source model we used a volumetric grid. Grid points had a spacing of 5 mm, resulting in 8799 dipoles covering the whole brain. Source-level power envelopes were computed from the source-projected complex signal:

$$X_{src}\left(t,f\right) = A\left(r,f\right) * X_{sens}\left(t,f\right)^2 \tag{3}$$

where $A$ denotes the spatial filter, $X_{src}$ the source-level power envelopes, and $X_{sens}$ the sensor-level analytic signal. The 5 mm spacing was chosen so as to allow for an integer spatial downsampling of the Brainnetome atlas (*Fan et al., 2016*) which we used to map cortical (and subcortical) regions into 246 distinct areas.

## MI between pupil and spectral activity (sensor space)

To quantify global pupil-power coupling in sensor space, we calculated MI between frequency-specific power envelopes and pupil time traces. To this end, pupil time series were downsampled to the temporal resolution of the respective power envelope, and segments containing artifacts in power envelopes were also removed from the pupil traces. After normalizing both time series by means of a Gaussian-copula-based approach (*Ince et al., 2017*), MI was computed per frequency and MEG sensor.

Observed MI was evaluated against permutation distributions of surrogate MIs based on randomly time-shifting pupil traces 200 times, while avoiding a window of ±10 s around the match of both original time series. Using the log-transformed observed MI, and the mean and standard deviation of equally log-transformed permutation distributions, we computed z-values for each frequency and sensor. In a final step, these z-values were pooled across sensors and their mean across participants tested against zero, for each recording site separately. Test results are reported for uncorrected and FDR-corrected thresholds in *Figure 2*. Topographies in *Figure 2* show observed standardized MI values by sensor.

## Cross-correlation analysis (sensor space)

We tested for frequency-specific lags between sensor-level power envelopes and pupil time series (and their derivative) by means of cross-correlation with a maximum lag of 10 s. Cross-correlations were computed for each sensor and frequency, separately, after downsampling pupil time series to the respective frequency-specific temporal resolution. In order to allow for the comparison across MEG recording sites (with different sensor layouts), we first ordered the recording channels into 39 equally sized, non-overlapping spatial bins, sorted from anterior to posterior channels. We next averaged the cross-correlograms per bin and additionally averaged the cross-correlations across three distinct frequency ranges: 2–4, 8–16, and 64–128 Hz. Due to the varying width of the wavelets for each center frequency, the temporal resolution, and, therefore, the number of samples, varied across frequencies. Hence, prior to averaging, we linearly interpolated the frequency-specific cross-correlograms such that the number of samples was equal to the number of samples of the highest frequency of each band (i.e., 4, 16, and 128 Hz).

## Source-level mapping of pupil-power correlations

To map pupil-power associations across each of the 246 cortical regions of the Brainnetome atlas, we correlated the forward-shifted (0.93 s) pupil time series with frequency-specific power envelopes by means of Spearman's rank correlation for each of the 8799 voxels of the source model, separately, after downsampling pupil time series to the respective frequency-specific temporal resolution.

All steps were repeated for correlations between pupil derivative and power fluctuations, however without accounting for the lag as previous research showed near-instantaneous correlations between neuromodulatory activity and fluctuations in pupil derivative (*Reimer et al., 2016*).

## Source-level power spectra and spectral parametrization

To correlate fluctuations in pupil diameter (and its temporal derivative) with fluctuations in the slope of the aperiodic component of the power spectra (henceforth referred to as the 'spectral exponent'), we first computed source-level power spectra. To this end, we source-projected the broad-band signal using linear beamforming (LCMV; *Van Veen et al., 1997*) instead of DICS (*Gross et al., 2001*). The procedure was similar to DICS, with the difference that the spatial filters were obtained using the cross-spectral density matrix averaged across all 25 frequency bands of interest. In keeping with the previous analysis, we used a regularization parameter of 5%.

Next, we divided the resulting source-level broad-band signal into non-overlapping temporal segments of 2 s length. For each segment, power spectra were computed using MATLAB's 'pwelch' function, ranging from 2 to 128 Hz, with a frequency resolution of 0.5 Hz. In addition, for each temporal segment, the mean pupil diameter as well as the mean of the pupil derivative were computed after shifting the pupil diameter, but not the pupil-derivative signal forward by 930 ms relative to the MEG time. Note that we opted for a different spectral decomposition method here than the wavelet-based approach (as described above) for practical reasons: Welch's periodogram method yields a direct estimate of the spectrum, which is convenient for parametrization as this analysis focused on the 1/f feature of the power spectrum rather than on power at any individual frequency.

To separate aperiodic and periodic components of the power spectra, we used the spectral parameterization toolbox 'FOOOF' for Python 3.7 (*Donoghue et al., 2020*). The algorithm iteratively fits neuronal power spectra as the sum of a Lorentzian function and a variable number of Gaussians of varying width, height, and mean (corresponding to the center frequency). To constrain the fitting procedure, we limited the number of periodic components to six and defined a minimum peak height of 0.05, with the minimum and maximum bandwidth of each peak set to 1 and 8 Hz. The knee of the Lorentzian was set to zero.

The algorithm was applied to the power spectra of each individual temporal segment (see above), which provided time-resolved estimates of periodic and aperiodic components. While the slope of the aperiodic component is relatively stable for lower frequencies up until ~40–50 Hz, the rate at which power changes can differ substantially for higher frequencies. This means that fitting power spectra uniformly across the entire frequency range may result in poorer fits and, consequently, misleading parameter estimates. Thus, and largely consistent with previous work (*Pfeffer et al., 2021*; *Waschke et al., 2021*), neuronal power spectra were fitted in the frequency range from 3 to 40 Hz.

## Polynomial model

We quantified quadratic components of the relation between pupil diameter (and pupil derivative) and band-limited activity fluctuations in spectral power. To this end, we sorted the aforementioned temporal segments (see section above: 'Source-level power spectra and spectral parametrization') based on pupil diameter (or pupil derivative) into 14 non-overlapping and equidistant bins. Next, we computed the average power for each of the bins and normalized (z-scored) the obtained values. Quadratic relationships were quantified by fitting the power vs. pupil (or pupil derivative) function with a polynomial of degree 2, separately for each participant, recording block (Hamburg only) and each of the 25 MEG center frequencies:

$$P_{src}\left(s,f\right) = \beta_0 + \beta_1 x + \beta_2 x^2 \tag{4}$$

where $s$ denotes the source location (with $s$ being 1–246 regions of the Brainnetome atlas), $f$ is the MEG center frequency (from 2 to 128 Hz, in 25 logarithmic steps), and $x$ is the mean pupil diameter for each pupil bin. The coefficients for the quadratic and the linear component are denoted with $\beta_2$ and $\beta_1$, respectively, and $\beta_0$ describes the offset. The polynomial was fitted by minimizing the sum of the squares of the difference between the polynomial model and the power vs. pupil (pupil derivative) functions (as implemented in MATLAB's 'polyfit' function). The p-values shown in *Figure 5* and *Figure 5—figure supplement 1* were obtained by testing the obtained beta-weights ($\beta_2$) against zero on the group level (two-tailed paired t-test).

## Acknowledgements

TP and THD thank Christiane Reissmann and Karin Deazle for assistance with MEG recordings and the acquisition of structural brain scans. CK, AK, GT, and JG thank Gavin Paterson and Frances Crabbe for expert assistance with MEG recordings and structural brain scans. CK thanks Linbi Hong, Josef Faller, and Paul Sajda for comments on the research. DSK and JG thank Hildegard Deitermann, Ute Trompeter, and Karin Wilken for assistance with MEG recordings and structural brain scans. All authors thank Ruud L van den Brink for helpful comments on the manuscript. Funding TP has been supported by a Feodor-Lynen fellowship of the Alexander-von-Humboldt Foundation. CK and AK have been supported by a Wellcome Trust Senior Investigator Grant awarded to JG (#098433) and GT (#98434). CK received further support through BBSRC Flexible Talent Mobility funding (BB/R506576/1) awarded by the University of Glasgow. THD has been supported by the German Research Foundation (DFG, grants DO 1240/3–1, DO 1240/4–1, and SFB 936, project numbers A7 and Z3) and the German Federal Ministry of Education and Research (BMBF, grants 01GQ1907 and 01EW2007B). JG has been supported by the Interdisciplinary Center for Clinical Research (IZKF) of the medical faculty of Münster (grant number Gro3/001/19) and the DFG (GR 2024/5-1)

## Additional information

### Competing interests

Tobias H Donner: Reviewing editor, eLife. The other authors declare that no competing interests exist.

### Funding

| Funder | Grant reference number | Author |
|---|---|---|
| Alexander von Humboldt-Stiftung | Feodor-Lynen Fellowship | Thomas Pfeffer |
| Wellcome Trust | Senior Investigator Grant #098433 | Joachim Gross |
| Wellcome Trust | Senior Investigator Grant #98434 | Gregor Thut |
| University of Glasgow | BBSRC Flexible Talent Mobility funding (BB/R506576/1) | Christian Keitel |
| Deutsche Forschungsgemeinschaft | DO 1240/3-1 | Tobias H Donner |
| Deutsche Forschungsgemeinschaft | DO 1240/4-1 | Tobias H Donner |
| Deutsche Forschungsgemeinschaft | SFB 936 A7/Z3 | Tobias H Donner |
| Bundesministerium für Bildung und Forschung | 01GQ1907 | Tobias H Donner |
| Bundesministerium für Bildung und Forschung | 01EW2007B | Tobias H Donner |
| Interdisciplinary Center for Clinical Research (IZKF) of the Medical Faculty of Münster | Gro3/001/19 | Joachim Gross |
| Deutsche Forschungsgemeinschaft | GR 2024/5-1 | Joachim Gross |

The funders had no role in study design, data collection and interpretation, or the decision to submit the work for publication.

## Author contributions
Thomas Pfeffer, Conceptualization, Data curation, Formal analysis, Funding acquisition, Investigation, Methodology, Project administration, Software, Validation, Visualization, Writing – original draft, Writing – review and editing; Christian Keitel, Conceptualization, Data curation, Formal analysis, Investigation, Methodology, Project administration, Software, Validation, Visualization, Writing – original draft, Writing – review and editing; Daniel S Kluger, Data curation, Formal analysis, Investigation, Validation, Writing – review and editing; Anne Keitel, Investigation, Validation, Writing – review and editing; Alena Russmann, Conceptualization, Formal analysis, Writing – review and editing; Gregor Thut, Funding acquisition, Resources, Writing – review and editing; Tobias H Donner, Conceptualization, Funding acquisition, Methodology, Resources, Shared senior author, Supervision, Writing – review and editing; Joachim Gross, Conceptualization, Funding acquisition, Methodology, Resources, Supervision, Writing – review and editing

## Author ORCIDs
Thomas Pfeffer http://orcid.org/0000-0001-9561-3085
Christian Keitel http://orcid.org/0000-0003-2597-5499
Daniel S Kluger http://orcid.org/0000-0002-0691-794X
Anne Keitel http://orcid.org/0000-0003-4498-0146
Tobias H Donner http://orcid.org/0000-0002-7559-6019
Joachim Gross http://orcid.org/0000-0002-3994-1006

## Ethics
Human subjects: Human subjects were recruited and participated in the experiments in accordance with the ethics committee responsible for the University Medical Center Hamburg-Eppendorf (Hamburg MEG data) approval number PV4648, the ethics committee of the University of Glasgow, College of Science and Engineering (Glasgow MEG data) approval number 300140078, and the ethics committee of the University of Muenster (Muenster MEG data) approval number 2018-068-f-S. All participants gave written informed consent prior to all experimental procedures and received monetary compensation for their participation.

## Decision letter and Author response
Decision letter https://doi.org/10.7554/eLife.71890.sa1
Author response https://doi.org/10.7554/eLife.71890.sa2

---

# Additional files

## Supplementary files
• Transparent reporting form

## Data availability
The ethics protocol(s) disallow sharing raw and preprocessed MEG and MRI data via a public repository. Data may be shared however within the context of a collaboration. No proposal is needed. However, the results presented in the manuscript are based on three separate datasets, collected independently in three different laboratories. As such, in order to obtain the data, an (informal) email to the authors responsible for the respective data sets is required (Hamburg: Thomas Pfeffer, thms.pfffr@gmail.com; Glasgow: Anne Keitel, a.keitel@dundee.ac.uk; Münster: Daniel Kluger, daniel.kluger@wwu.de). The code and data immediately underlying all main and supplementary figures has been made publicly available. Source data has been uploaded to a public repository (https://osf.io/fw4bt), along with MATLAB code that was used to generate the main and supplementary figures.

The following dataset was generated:

| Author(s) | Year | Dataset title | Dataset URL | Database and Identifier |
|---|---|---|---|---|
| Thomas Pfeffer, Christian Keitel, Daniel Kluger, Anne Keitel, Gregor Thuth, Tobias H. Donner, Joachim Gross | 2022 | Code and data accompanying Pfeffer, Keitel et al. (2021). Coupling of pupil- and neuronal population dynamics reveals diverse influences of arousal on cortical processing | https://osf.io/fw4bt | Open Science Framework, fw4bt |

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
