## [Editor Report]

The authors have mapped how fluctuations of pupil-linked arousal and cortical activity co-vary in the human brain at rest recorded using MEG. This was achieved in a large sample of participants (N=81), and the results reveal diverse and consistent arousal effects on band-limited cortical activity. These findings provide important insight into how subcortical activity associated with arousal is reflected in neocortical dynamics.

---

## [Decision Letter]

**Decision letter after peer review:**

Thank you for submitting your article "Coupling of pupil-and neuronal population dynamics reveals diverse influences of arousal on cortical processing" for consideration by *eLife*. Your article has been reviewed by 3 peer reviewers, and the evaluation has been overseen by a Reviewing Editor and Laura Colgin as the Senior Editor. The following individuals involved in review of your submission have agreed to reveal their identity: Martin Dahl (Reviewer #1); Tzvetan Popov (Reviewer #2); Andrew J Quinn (Reviewer #3).

Essential revisions:

In general, the referees are very positive, but further clarifications are needed for some of the analysis aspects as well as the embedding in the existing literature. The referees have made a set of constructive suggestions. Some of the overarching points being:

1) Methodological issues- e.g.

– The applied time-lag between pupil size and the MEG signal.

– Morlet vs sliding time-window

2) Relationship between eye movements and pupil dilation.

3) Discussion on E:I balance.

4) Relation to the existing literature on fMRI and animals research.

*Reviewer #1 (Recommendations for the authors):*

1. Please define the term arousal before usage. This definition is crucial, as previous research defined arousal based on behavioral outcomes (e.g., "[…], for the purposes of this review, the term arousal refers to a continuum of sensitivity to environmental stimuli." [15]) and tied it closely to alterations in cortical EEG synchronization ("Alterations in arousal state are accompanied by alterations in forebrain neuronal activity, reflected in electroencephalographic (EEG) signals" [15], referring to work by Moruzzi and Magoun, 1949 [16]).

Behavioral responses are not collected in this study. Defining arousal based on M/EEG responses does not seem to work here either (as the authors attempt to demonstrate this association). Is arousal equated with pupil-indexed neuromodulation here (l. 45)?

Related: The authors introduce pupil diameters as proxy for arousal and index of cortical state. Are these two terms (arousal state, cortical state) referring to the same or different things?

2. The authors should mention at some point that also serotonergic[17] and even dopaminergic[18] neuromodulation has been related to pupil dilation.

3. I find the methods section somewhat difficult to read, likely due to the three-site design. For different sites, varying aspects of data collection and processing are reported. For instance, only for the Glasgow sample it is mentioned how the head shape was acquired – was this not done for the Hamburg and Münster samples? I suggest including a table directly contrasting the collection and preprocessing steps for the different sites. Figure S1 does indicate some site effects and the suggested table may help readers to guess what may be the underlying cause.

4. How did the authors determine the cut-off frequencies for their pupil band-pass filter? Previous research demonstrated (task-related) pupil fluctuations at 2 Hz [19], thus the upper limit does seem somewhat low.

5. The methods indicate that no time periods were removed due to invalid pupil data (e.g., l. 785 following). Please provide more information concerning the pupil segments that were interpolated? (mean, SD; range; % of raw data)

6. At some points, the methods could benefit from more details. To give only a few examples: The estimation of the Spearman's correlation in source space is only reported in the results but not the methods. Moreover (~l. 840) the Methods do not specify that statistics are run within participants and ß parameters are tested against zero on a group level. Finally, (~l. 800): The authors could state how they computed the cross-correlation (e.g., what is the max lag?) before reporting how they averaged the cross-correlations across sensors and frequencies.

7. ~l. 170 "at a lag closer to zero (210 ms; negative peak)" – shouldn't this be a positive peak for the α-β band?

8. "l.265 As anterior cingulate cortex, a region receiving strong innervations from locus coeruleus (Chandler et al., 2013)." It could be added here that AAC also projects to the LC, thus, there are bi-directional connections [2,20]

9. l. 315"Above described analyses focused on quantifying linear relationships between intrinsic fluctuations in pupil diameter, as well as pupil-derivative, and band-limited cortical population activity." Should be monotonic, not linear?

10. L. 395 Here statistics are reported in full in the main text, but for parts of the paper they are not (e.g., Figure 5 A). I think the authors should be consistent and also report the other statistics.

11. l. 480 "More indirect multi-synaptic route to neuromodulation of occipital α activity may delay the lag relative to other effects and put it closer to the time of maximum pupil dilation." While this argument seems intuitive, I was wondering whether the authors can provide any empirical support for a differential delay of direct cortical and thalamus-mediated α-β changes due to neuromodulation.

12. l. 560 "More specifically, in the case of noradrenaline, the activation of high-affinity α2 receptors at medium arousal level may (directly or indirectly) lead to increased α- and β-band activity" This reference seems suitable here [9]

13. The hippus findings do not seem to be reported in the manuscript? (only a Figure in the SI)

Cited references:

1 Reimer, J. et al. (2016) Pupil fluctuations track rapid changes in adrenergic and cholinergic activity in cortex. Nat. Commun. 7, 13289

2 Joshi, S. et al. (2016) Relationships between pupil diameter and neuronal activity in the locus coeruleus, colliculi, and cingulate cortex. Neuron 89, 221-234

3 Kucyi, A. and Parvizi, J. (2020) Pupillary dynamics link spontaneous and task-evoked activations recorded directly from human insula. J. Neurosci. 40, 6207-6218

4 Dahl, M.J. et al. (2020) Noradrenergic responsiveness supports selective attention across the adult lifespan. J. Neurosci. 40, 4372-4390

5 Kosciessa, J.Q. et al. (2021) Thalamocortical excitability modulation guides human perception under uncertainty. Nat. Commun. 12, 1-15

6 Whitmarsh, S. et al. (2021) Neuronal correlates of the subjective experience of attention. Eur. J. Neurosci. DOI: 10.1111/ejn.15395

7 Nestvogel, D.B. and Mccormick, D.A. (2021) Visual Thalamocortical Mechanisms of Waking State Dependent Activity and Α Oscillations. bioRxiv DOI: 10.1101/2021.04.14.439865

8 Senzai, Y. et al. (2019) Layer-Specific Physiological Features and Interlaminar Interactions in the Primary Visual Cortex of the Mouse. Neuron 101, 500-513.e5

9 Buzsáki, G. et al. (1991) Noradrenergic control of thalamic oscillation: The role of α‐2 receptors. Eur. J. Neurosci. 3, 222-229

10 Buzsáki, G. et al. (1988) Nucleus basalis and thalamic control of neocortical activity in the freely moving rat. J. Neurosci. 8, 4007-26

11 McCormick, D.A. (1989) Cholinergic and noradrenergic modulation of thalamocortical processing. Trends Neurosci. 12, 215-221

12 Goard, M. and Dan, Y. (2009) Basal forebrain activation enhances cortical coding of natural scenes. Nat. Neurosci. 12, 1444-1449

13 Rodenkirch, C. et al. (2019) Locus coeruleus activation enhances thalamic feature selectivity via norepinephrine regulation of intrathalamic circuit dynamics. Nat. Neurosci. 22, 120-133

14 Totah, N.K. et al. (2021) Synchronous spiking associated with prefrontal high γ oscillations evokes a 5 Hz-rhythmic modulation of spiking in locus coeruleus. J. Neurophysiol. DOI: 10.1152/jn.00677.2020

15 Berridge, C.W. (2008) Noradrenergic modulation of arousal. Brain Res. Rev. 58, 1-17

16 Moruzzi, G. and Magoun, H.W.W. (1949) Brain stem reticular formation and activation of the EEG. Electroencephalogr. Clin. Neurophysiol. 1, 455-473

17 Cazettes, F. et al. (2020) Phasic Activation of Dorsal Raphe Serotonergic Neurons Increases Pupil Size. Curr. Biol. 0,

18 de Gee, J.W. et al. (2017) Dynamic modulation of decision biases by brainstem arousal systems. e*Life* 6,

19 Schwiedrzik, C.M. and Sudmann, S.S. (2020) Pupil diameter tracks statistical structure in the environment to increase visual sensitivity. J. Neurosci. 40,

20 Aston-Jones, G. and Cohen, J.D. (2005) An integrative theory of locus coeruleus-norepinephrine function: Adaptive gain and optimal performance. Annu. Rev. Neurosci. 28, 403-450

*Reviewer #2 (Recommendations for the authors):*

This is a very strong paper; rationale and approach are well motivated and carried out.

Some issues came to mind.

1. Eye movements.

There is a clear relationship between eye movements and pupil dilation (e.g. https://peerj.com/articles/1150/). This is not only the case for large saccades but likely for fixational eye movements (FEM) too. Both are likely present during the recording. I wondered, could the authors extract time series of gaze position and use these as a control analysis instead of the pupil time series? Does pupil dilation vary with gaze direction eccentricity? Does gaze direction relate to brain oscillations too or not at all. I guess a simple check is to plot the x-y- ET data together with pupil diameter locked to saccade (both micro and macro) onset. Next, one could re-do the main analysis (e.g. Figure 2 and 5) on data segments pre movement vs. post movement followed by equivalence test or something.

In particular, the anterior δ-theta effects and the visual α could be interpreted within the context of eye movements too. In fact, this could be the one of the "independent mechanisms" the authors speculate about on p. 6 line 216. The authors note that the effect in the visual cortices had the shortest lag with respect to pupil dilation. Yet, visual areas are early in the hierarchy of areas controlling and noticing eye movement. For example if one considers the observations that, cooling of V1 silences cell firing in the superior colliculus (i.e. extended work by Peter Shiller) and in turn eye movement, or some recent work on motor fields in V1 (https://science.sciencemag.org/content/370/6521/1191).

2. MEG gradiometer vs. magnetometer

Is there a difference in the correlation maps between the magnetometer and gradiometer data? A supplemental figure would help here. Basically Figure 4D split by side? Also, is the spectral exponent different for different sensor types? Magnetometers would be likely noisier hence a different aperiodic activity pattern may arise. If so does this impact the pooled results?

From Figure 2B it seems that the data was converted to planar gradient, if so it needs to be mentioned in the methods.

Were head movements monitored and accounted for? Particularly in the context of the data harmonization it would be informative for the reader in the method section. To be clear, the fact that the authors do find effects even if not corrected for head movements speaks for itself. Yet, this is an ongoing debate in MEG research and needs to be mentioned.

3. Pupil data. I was confused what the pupil data actually is. From p. 5 line 164 I understood that cross-correlations were computed between sensor level power and pupil (power? or size?). Later on in Methods the exponent of pupil power spectra is mentioned. All Figures suggest pupil size as indicated by the eye cartoon. Please specify what data (size, power, exponent) is used in which of the different analyses. E.g. Figure 6A is this a correlation between spectral exponent(brain) vs. spectral exponent (pupil)? Do the main results replicate if one uses the absolute pupil size?

RE: The E/I idea. Figure 6 implies then that the whole cortex is a state of excitation which by itself appears in contrast with functional specificity of the E/I idea. In fact, a bit against the authors interpretation on line 441 "modulations of specific cortical oscillations by arousal". All examined band widths and 1/f exponent relate to arousal. The "specificity" argument needs clarification.

Furthermore, I wonder to what extend the same is true if the authors redo the analysis on the planar gradient data? Can the beamformer weakness in modeling deep sources manifest in noise consistency which in turn is captured by the exponent correlation bit?

*Reviewer #3 (Recommendations for the authors):*

The following points relate to points in the public review

#1 spectral analyses. The two analysis types are:

Line 745 – "MEG spectral analysis (all sites)", describes a Morlet wavelet analyses which appears to be used for the results in figures 1-4.

Line 325 and 817 – "Nonlinear relations between pupil-linked arousal and band-limited cortical activity", describes a two second sliding window pwelch analysis.

The sliding window Welch's method analysis will have the same temporal resolution for each frequency band, and the pupil dynamics are matched to the sliding windows used for the MEG. I would strongly suggest running both analyses with Welch's periodogram across sliding windows. This would ensure that dynamics in all frequency have the same resolution and that all analyses in the paper are matched. If there is an additional concern that successive window are independent for the polynomial modelling then the window overlap could be adjusted between the two sections (though independent windows is potentially desirable for the first analysis as well?)

#2 An additional control analysis exploring whether the spatial correlation maps in are associated with source power in the presented frequencies. For example, figure 5D shows the spatial map of the quadratic coefficient at 11 and 19Hz – how closly related is this topography to the simpler spatial distribution of 11Hz and 19Hz power?

#3 From the discussion in Gao et al.,: “Hence, we propose that slope changes in a particular frequency region (30-70 Hz) correspond to changes in E:I balance, while making no claims about other frequency regions”. Not sure what to recommend about the frequency range difference here – at the least it should be acknowledged and discussed. Ideally this analysis should be matched to Gao et al., given how critical this reference is to the discussion and interpretation.

Additional points

#4 It isn't clear where the three frequency bands of interest (line 188: 2-4Hz, 8-16Hz, and 64-128Hz) come from. I think this is based on Figure 1 but only the Glasgow site appears to show distinct clusters in these bands. Hamburg and Munster are significant across the whole 2-32Hz range and Hamburg is additionally significant from 32-128Hz. On this basis, the exclusion of the 4-8Hz and 16-64Hz range seems a bit odd. Some additional clarification and justification would be helpful – apologies if I have missed this in another section.

#5 related to 4, subjectively the MI spectra across the three sites don't seem to be all that similar, a stronger justification for combining subjects in subsequent analyses would be reassuring. Given that this is the jumping off point for the rest of the paper it would be good to revisit this figure to emphasise the points of similarity – the supporting spectra in supplemental materials and topos show between site similarity more convincingly.

To formalise this, can you repeat the between subject correlation matrix from figure S1B for the MI spectra and see if there is substantial banding per-site? if site-specific bands are visible here, then a mixed-modelling approach or the inclusion of a site-covariate in subsequent analyses would be desirable.

---

## [Author Response]

Essential revisions:In general, the referees are very positive, but further clarifications are needed for some of the analysis aspects as well as the embedding in the existing literature. The referees have made a set of constructive suggestions. Some of the overarching points being:1) Methodological issues- e.g.– The applied time-lag between pupil size and the MEG signal.– Morlet vs sliding time-window2) Relationship between eye movements and pupil dilation.3) Discussion on E:I balance.4) Relation to the existing literature on fMRI and animals research.

We have addressed all issues with a focus on those listed above. Please find our point-by-point responses to the Reviewer comments below.

We would like to highlight that we found a minor error in how p-values, specifically the FDR correction was reported in the paper. In the process of revising, and acknowledging a point raised by Reviewer 1, we have corrected this error while unifying the approach. Specifically, we now use, and report: (1) a threshold of q=0.1 for all FDR corrections carried out in the paper, and (2) specific FDR-adjusted α levels. Importantly, this unification led to minor quantitative changes in results and illustrations but did not change any of our qualitative conclusions.

Note that all publications we have referenced in this response letter are cited in the manuscript.

Reviewer #1 (Recommendations for the authors):1. Please define the term arousal before usage. This definition is crucial, as previous research defined arousal based on behavioral outcomes (e.g., "[…], for the purposes of this review, the term arousal refers to a continuum of sensitivity to environmental stimuli." [15]) and tied it closely to alterations in cortical EEG synchronization ("Alterations in arousal state are accompanied by alterations in forebrain neuronal activity, reflected in electroencephalographic (EEG) signals" [15], referring to work by Moruzzi and Magoun, 1949 [16]).Behavioral responses are not collected in this study. Defining arousal based on M/EEG responses does not seem to work here either (as the authors attempt to demonstrate this association). Is arousal equated with pupil-indexed neuromodulation here (l. 45)?

Yes. We are using the term “*pupil-linked arousal*” frequently to make this point. Please note that we emphasise the link between arousal and neuromodulation early:

“Fluctuations in arousal, controlled by subcortical neuromodulatory systems, continuously shape cortical state, with profound consequences for information processing.” (first sentence of the abstract).

From our perspective, this provides a strong neuromodulatory angle on arousal from the outset. Following your comment we have reviewed language throughout for consistency.

Related: The authors introduce pupil diameters as proxy for arousal and index of cortical state. Are these two terms (arousal state, cortical state) referring to the same or different things?

We refer to different things: We use pupil dilations as a proxy for arousal and want to know how this co-varies with “cortical state” as indexed by MEG-recorded activity (periodic and aperiodic). This is captured in panel B of our Figure 1. We have modified the two mentions of “arousal state” to just “arousal” and added wording to the following sentence in the introduction to disambiguate:

“Pupil diameter has recently attracted interest in neuroscience as a peripheral index of how arousal influences cortical state (McGinley, David, et al., 2015; Reimer et al., 2014; Vinck et al., 2015).”

2. The authors should mention at some point that also serotonergic[17] and even dopaminergic[18] neuromodulation has been related to pupil dilation.

We agree that this is important. However, it is difficult to assess to what extent pupil dilations are a direct consequence of serotonergic or dopaminergic activity or merely a consequence of strong interconnectivity between neuromodulatory nuclei, especially by projections from locus coeruleus. For instance, in their Discussion, Cazettes et al., (reference [17]) report latencies between dorsal raphe (DR) stimulation and pupil dilation that are ~100 ms longer than pupil dilations elicited through stimulation of noradrenergic neurons (see their Discussion), which supports the idea of DR effects on pupil being mediated by locus coeruleus. Moreover, Breton-Provencher and Sur (2019) assessed the direct causal contribution of the locus coeruleus to the arousal-related (sound-induced) pupil response, showing that the silencing of noradrenergic neurons in LC abolished (or strongly attenuated) the pupil response to arousing auditory stimuli (see their Supplementary Figure 8).

Nonetheless, we agree with this point. Therefore, we have added the following information to the first paragraph of the section ‘Arousal modulates cortical activity across space, time and frequencies’ in the Discussion as follows:

“Additionally, this may elucidate the roles of serotonergic and dopaminergic

Neuromodulation, which also influence pupil size (Cazettes et al., 2021; de Gee et al., 2017).”

3. I find the methods section somewhat difficult to read, likely due to the three-site design. For different sites, varying aspects of data collection and processing are reported. For instance, only for the Glasgow sample it is mentioned how the head shape was acquired – was this not done for the Hamburg and Münster samples? I suggest including a table directly contrasting the collection and preprocessing steps for the different sites. Figure S1 does indicate some site effects and the suggested table may help readers to guess what may be the underlying cause.

We have revised the Methods section to align the data acquisition reporting between the three labs more strongly, while allowing for an evaluation of potential influences that would explain (non-consequential) differences. In the process, we have added details on headshape acquisition in the lab-specific ‘Participants and Data Acquisition’ sections. We have refrained from adding the suggested table as we felt that it would be difficult to contrast all differences between sites exhaustively, and in a concise and comprehensible fashion.

4. How did the authors determine the cut-off frequencies for their pupil band-pass filter? Previous research demonstrated (task-related) pupil fluctuations at 2 Hz [19], thus the upper limit does seem somewhat low.

We set the low-pass cut-off at 2 Hz as this was the lowest frequency that we analysed with respect to power fluctuations in cortical oscillations. Also, it does capture the time scales of arousal-linked pupillary phenomena such as the Hippus (~0.2 Hz) and the canonical response to arousing events (< 1 Hz given an average t_max_ of 930msec). Further, data of McGinley et al., (2015) and Reimer et al., (2016) suggests that neuromodulation has the most profound influences on pupil dynamics below < 1 Hz. We have added these justifications to the Methods section.

It is worth pointing out that the suggested journal article [19] investigates pupil “entrainment” to physical changes in visual input, a response that depends on different neural circuitry and may therefore underlie different time scales than the arousal-linked changes of interest here.

5. The methods indicate that no time periods were removed due to invalid pupil data (e.g., l. 785 following). Please provide more information concerning the pupil segments that were interpolated? (mean, SD; range; % of raw data)

We have added the following data to the Methods section:

“The number of blinks and, therefore, the percentage of the interpolated data points varied across MEG sites: in total, 17.9% of the Glasgow pupil time series (standard deviation: 17.9%; range: 1.3 – 39.8%, with one outlier of 83.3%), 11.9% of the Hamburg pupil time series (standard deviation: range; 11.3%; 0 – 37.1%) and 22.9% of the Münster pupil time series (standard deviation: 14.9%; range: 0 – 56.7%) were interpolated.”

6. At some points, the methods could benefit from more details. To give only a few examples: The estimation of the Spearman's correlation in source space is only reported in the results but not the methods. Moreover (~l. 840) the Methods do not specify that statistics are run within participants and ß parameters are tested against zero on a group level. Finally, (~l. 800): The authors could state how they computed the cross-correlation (e.g., what is the max lag?) before reporting how they averaged the cross-correlations across sensors and frequencies.

We have added the requested details to the Methods section and revised that section further to include changes made in response to the other Reviewers’ comments.

With regard to the statistics on the ß parameters, this may be a misunderstanding: the statistics are not run within participants, but indeed only tested against zero on a group level. The polynomial model is fitted within subjects, in order to obtain parameters for each subject and block (only for the Hamburg data where two blocks were recorded). The statistics were then run by testing the ß parameters against zero on the group level. No statistics were obtained from running within-subjects tests. We now mention this more explicitly in the Methods section of the revised manuscript.

7. ~l. 170 "at a lag closer to zero (210 ms; negative peak)" – shouldn't this be a positive peak for the α-β band?

Yes, has been corrected. Thanks for the catch!

8. "l.265 As anterior cingulate cortex, a region receiving strong innervations from locus coeruleus (Chandler et al., 2013)." It could be added here that AAC also projects to the LC, thus, there are bi-directional connections [2,20]

We have added a note and the references on bi-directional connections to this sentence

(Results section ‘Spatial and spectral dissociations of pupil-power correlations’):

“[…] as well as anterior cingulate cortex, a region that receives strong innervations from the locus coeruleus (Chandler
et
al.,
2013), while also projecting to the LC itself (Joshi et al., 2016; Aston-Jones and Cohen, 2005).”

9. l. 315"Above described analyses focused on quantifying linear relationships between intrinsic fluctuations in pupil diameter, as well as pupil-derivative, and band-limited cortical population activity." Should be monotonic, not linear?

Following your comment, we have changed ‘linear’ to ‘monotonic’ where applicable to be inclusive of results based on rank correlations (as used for the mapping in Figure 4). Whenever we use ‘non-linear’ this is with explicit reference to the quadratic fit or inverted-U shape to avoid confusion with monotonous yet non-linear relationships.

10. L. 395 Here statistics are reported in full in the main text, but for parts of the paper they are not (e.g., Figure 5 A). I think the authors should be consistent and also report the other statistics.

Note that the case of L395 constitutes a special situation where the analysis approach resulted in a small number of tests that we were able to report directly. Most other statistical comparisons have been carried out on higher dimensional data (across space, frequencies, time or a combination of those) making it difficult to give a concise description in the text.

Statistical results are therefore largely reported in the graphs.

11. l. 480 "More indirect multi-synaptic route to neuromodulation of occipital α activity may delay the lag relative to other effects and put it closer to the time of maximum pupil dilation." While this argument seems intuitive, I was wondering whether the authors can provide any empirical support for a differential delay of direct cortical and thalamus-mediated α-β changes due to neuromodulation.

The sentence in question rephrases the previous sentence that also provides empirical support. We have extended this section (Discussion, ‘Arousal modulates cortical activity across space, time and frequencies’), also incorporating some of the suggested literature to clarify as follows:

“This short-lag positive correlation may stem from interactions between the locus coeruleus and regions of the thalamus (McCormick, 1989; McCormick et al., 1991; Stitt et al., 2018), in particular those implicated in the generation of occipital α-band activity via thalamo-cortical loops (e.g. the Pulvinar; Saalmann et al., 2012). These loops may therefore be sensitive to noradrenergic (Dahl et al., 2022) or even basal-forebrain regulated cholinergic modulation (Nestvogel and McCormick, 2021; Zagha and McCormick, 2014). Both scenarios posit indirect multi-synaptic routes to the neuromodulation of occipital α activity that may delay the lag relative to other effects and put it closer to the time of maximum pupil dilation.”

12. l. 560 "More specifically, in the case of noradrenaline, the activation of high-affinity α2 receptors at medium arousal level may (directly or indirectly) lead to increased α- and β-band activity" This reference seems suitable here [9]

Reference has been added.

13. The hippus findings do not seem to be reported in the manuscript? (only a Figure in the SI)

Yes. Our main aim was to describe the relationship between cortical activity and pupil-linked arousal, rather than characterising the spectral make up of either signal alone. We present these findings in the SI to support our pooling of data across all sites. The

Hippus is taken as one benchmark to demonstrate the similarity of all datasets.

Reviewer #2 (Recommendations for the authors):This is a very strong paper; rationale and approach are well motivated and carried out.Some issues came to mind.

Thank you for your positive and constructive feedback*.*

1. Eye movements.There is a clear relationship between eye movements and pupil dilation (e.g. https://peerj.com/articles/1150/). This is not only the case for large saccades but likely for fixational eye movements (FEM) too. Both are likely present during the recording. I wondered, could the authors extract time series of gaze position and use these as a control analysis instead of the pupil time series? Does pupil dilation vary with gaze direction eccentricity? Does gaze direction relate to brain oscillations too or not at all. I guess a simple check is to plot the x-y- ET data together with pupil diameter locked to saccade (both micro and macro) onset. Next, one could re-do the main analysis (e.g. Figure 2 and 5) on data segments pre movement vs. post movement followed by equivalence test or something.In particular, the anterior δ-theta effects and the visual α could be interpreted within the context of eye movements too. In fact, this could be the one of the "independent mechanisms" the authors speculate about on p. 6 line 216. The authors note that the effect in the visual cortices had the shortest lag with respect to pupil dilation. Yet, visual areas are early in the hierarchy of areas controlling and noticing eye movement. For example if one considers the observations that, cooling of V1 silences cell firing in the superior colliculus (i.e. extended work by Peter Shiller) and in turn eye movement, or some recent work on motor fields in V1 (https://science.sciencemag.org/content/370/6521/1191).

We agree that controlling for eye movements is important. For this reason, we have regressed out the effect of (large) saccades on pupil diameter already during the preprocessing of the pupil time series (see Methods, section ‘Pupil preprocessing (all sites)’). All analyses reported in the manuscript are therefore unaffected by saccades.

In response to your comment, we have carried out several analyses in order to make sure that eye movements in general and microsaccades in particular do not explain our findings. To this end, we correlated gaze direction eccentricity with pupil diameter and analyzed microsaccade-locked time-frequency representations as well as changes in pupil diameter. Fluctuations in gaze eccentricity were not significantly correlated with fluctuations in pupil diameter (Mean correlation across subjects: r = -0.0158; P = 0.53; paired two-sided t-test). Next, we identified microsaccades using the Engbert-Kliegl algorithm (Engbert et al., 2003, Vis Res) with default parameters (minimal saccade duration of 5 samples, i.e., 12.5 msec and λ=6). For 10 subjects (Hamburg data), only horizontal eye movement data were saved. These subjects were excluded from the analysis. Furthermore, one subject of the Muenster dataset was excluded due to noisy gaze position data, resulting in 62 total subjects. Next, we computed event-locked time-frequency representations (sensor-level only). We found a significant reduction in α power following microsaccades (Figure 2—figure supplement 1A) extending from anterior to posterior sensors (Figure 2—figure supplement 1B). Critically, however, across all frequencies, there were no significant correlations between the change in pupil diameter and the change in cortical activity following microsaccades (Figure 2—figure supplement 1C). Moreover, pupil diameter does not change significantly following microsaccade onset (Figure 2—figure supplement 1D). Hence, while microsaccades seem to affect band-limited brain activity (across almost all sensors and most prominently in the α-range), they do not affect pupil diameter and the interaction of pupil diameter with cortical activity. Thus, we assume that microsaccades do not affect the correlations and results that are central to this paper.

We have added this control analysis to the manuscript as Figure 2—figure supplement 1 along with the following paragraph to the Results section (the procedure is described in detail in a new paragraph in the Methods section):

“A possible confound in identifying relationships between pupil and cortical dynamics are eye movements: Saccades change pupil size (Mathot, 2018; Mathôt et al., 2015) and oculomotor behaviour is functionally linked to cortical α oscillations (Popov et al., 2021). Here, we controlled for saccade effects by regressing out canonical responses from pupil time series (as detailed in the Methods section; see e.g., Urai et al., 2017). An additional control analysis found that remaining micro-saccades (not captured by the regressing-out) led to a transient suppression of cortical activity in the 8-32 Hz range. However, this effect was not associated with any changes in pupil diameter (Figure 2—figure supplement 1).”

2. MEG gradiometer vs. magnetometerIs there a difference in the correlation maps between the magnetometer and gradiometer data? A supplemental figure would help here. Basically Figure 4D split by side? Also, is the spectral exponent different for different sensor types? Magnetometers would be likely noisier hence a different aperiodic activity pattern may arise. If so does this impact the pooled results?

As requested, we have plotted the pupil-power correlation maps split by MEG-site to allow for a comparison between sensor types (Author response image 1) . Source maps are highly convergent, irrespective of sensor type, with highly similar spatial patterns across all four frequency bands of interest. Given that we already provide Figures 2B, 3A as well as Figure 1—figure supplements 1 and 2 that allow for a comparison of the results for the different sensor types we have refrained from adding this as a further supplement.

**Author response image 1. sa2fig1:** Source maps of pupil-power correlations for four frequency bands of interest. Similar to Figure 4D, but separately for each site. (A) Glasgow (B) Hamburg and (C) Muenster. Shown are unthresholded maps.

Regarding the spectral exponent, crucial for our analysis is the change in the slope of the power spectrum (for different degrees of pupil diameter), which seems unaffected by the “base slope” across sites, as shown in figure Author response image 2: while the mean spectral exponent varies across MEG sites (lowest in the Hamburg data, highest in the Glasgow data), we find highly similar linear relations between spectral exponent and pupil bin, with pearson correlation coefficients ranging between -0.94 (Muenster data) and -0.98 (Hamburg data).

**Author response image 2. sa2fig2:** Spectral exponent as a function of mean pupil diameter, separated by MEG site.

From Figure 2B it seems that the data was converted to planar gradient, if so it needs to be mentioned in the methods.

These plots do not involve computing a planar gradient. The “focusedness” is a result of the correlation with the pupil timeseries.

Were head movements monitored and accounted for? Particularly in the context of the data harmonization it would be informative for the reader in the method section. To be clear, the fact that the authors do find effects even if not corrected for head movements speaks for itself. Yet, this is an ongoing debate in MEG research and needs to be mentioned.

We have added information on head position monitoring where it was not previously available in the lab-specific ‘Participants and Data Acquisition’ sections of the Methods section.

3. Pupil data. I was confused what the pupil data actually is. From p. 5 line 164 I understood that cross-correlations were computed between sensor level power and pupil (power? or size?). Later on in Methods the exponent of pupil power spectra is mentioned. All Figures suggest pupil size as indicated by the eye cartoon. Please specify what data (size, power, exponent) is used in which of the different analyses. E.g. Figure 6A is this a correlation between spectral exponent(brain) vs. spectral exponent (pupil)? Do the main results replicate if one uses the absolute pupil size?

All analyses always use pupil size time series. We have only produced pupil spectra, and extracted parameters from them, for Figure 1—figure supplement 2 (Figure S2 in previous version of the manuscript), i.e. for demonstrating the similarity of pupil size recordings between laboratories. Following your comment we have clarified this in the Results section where appropriate and added the following sentences to the description of spectral analyses of pupil time series to the Methods section:

“We conducted additional spectral analyses of pupil time series for the purpose of cross-laboratory comparisons only – see Figure 1—figure supplement 2. All analyses reported in the Results are based on the pupil-diameter time series (or their derivative).”

RE: The E/I idea. Figure 6 implies then that the whole cortex is a state of excitation which by itself appears in contrast with functional specificity of the E/I idea. In fact, a bit against the authors interpretation on line 441 "modulations of specific cortical oscillations by arousal". All examined band widths and 1/f exponent relate to arousal. The "specificity" argument needs clarification.

This point may be based on a misunderstanding: Figure 6 does not display E/I ratio itself but only the covariation of E/I (by proxy of spectral slope) with pupil diameter. This still allows different brain areas to have different dominance of E or I. The negative correlation throughout the cortex only tells us that regional E/I changes in a similar direction with changes in pupil size.

We have clarified the “specificity argument” in the Results section ‘Couplings between pupil-linked arousal and periodic components of cortical population activity are not induced by couplings with aperiodic components’ as follows:

“These control analyses confirm that the couplings between pupil (and pupil-derivative) and band-limited cortical activity reported in the preceding sections were genuine reflections of arousal-related modulations of periodic cortical activity with distinct effects for different frequency bands.”

Furthermore, I wonder to what extend the same is true if the authors redo the analysis on the planar gradient data? Can the beamformer weakness in modeling deep sources manifest in noise consistency which in turn is captured by the exponent correlation bit?

Data were not transformed to planar gradients in our analysis but we assume that the reviewer is referring to sensor space data in general. In any case, that should not explain a *stronger* correlation between slope and pupil size for deeper sources. If at all, a depth-noise increment should attenuate correlations between pupil size and spectral exponents and would not account for the isotropy in source space. Again, we would like to emphasize that it is not the spectral exponent per se, but the change in the spectral exponent for different pupil diameters that is crucial for this analysis. In the same vein, note that Beamformer coefficients are static. Even if the beamformer depth bias changes the spectral slope compared to superficial areas it would not do so as a function of pupil size.

Therefore, it can not lead to increased correlation. (Also see response to point 2.)

Reviewer #3 (Recommendations for the authors):The following points relate to points in the public review#1 spectral analyses. The two analysis types are:Line 745 – "MEG spectral analysis (all sites)", describes a Morlet wavelet analyses which appears to be used for the results in figures 1-4.Line 325 and 817 – "Nonlinear relations between pupil-linked arousal and band-limited cortical activity", describes a two second sliding window pwelch analysis.The sliding window Welch's method analysis will have the same temporal resolution for each frequency band, and the pupil dynamics are matched to the sliding windows used for the MEG. I would strongly suggest running both analyses with Welch's periodogram across sliding windows. This would ensure that dynamics in all frequency have the same resolution and that all analyses in the paper are matched. If there is an additional concern that successive window are independent for the polynomial modelling then the window overlap could be adjusted between the two sections (though independent windows is potentially desirable for the first analysis as well?)

The reviewer is correct that we have applied wavelet-based approaches to spectral decomposition for all analyses except for the tests of nonlinear (quadratic) and aperiodic (spectral slope) relationships with pupil size, which are based on 2-sec sliding window short-term Fourier transforms based on Welch’s method.

While we understand that homogenizing the spectral analysis approaches may benefit the comprehensibility of the paper, we would like to keep with our two-pronged solution for practical reasons. We are convinced that the reviewer will agree that both approaches are equally valid and, in effect, largely equivalent – differences can be found in how both approaches detail adjusting spectral and temporal resolutions. To illustrate the similarities, we have recomputed Figure 2 with a pwelch approach for your comparison, see Author response image 3:

**Author response image 3. sa2fig3:** Comparison of results presented in Figure 2, obtained via different spectral decomposition approaches. Upper panel: Wavelet-based approach as used in the paper. Lower panel: Short-term Fourier transform (STFT) based approach using Welch’s method (0.5 sec window length). Note that the STFT approach seems to impose a 1/f effect onto the results, while also reducing sensitivity in the higher frequency ranges.

The wavelet approach as applied here does indeed change the temporal resolution for each frequency – the number of samples for pupil-power correlations increases with frequency. However, varying the temporal resolution also naturally accommodates different spectral resolutions, i.e. wider bands for higher frequencies, which is prudent given our logarithmic frequency scale. In turn, the window length of a short-term Fourier transform (pwelch) based sliding window approach would be determined by the lowest frequency of interest, in our case 2 Hz, and result in a window length of 0.5 sec, with a corresponding fixed frequency resolution of 2 Hz. This resolution may befit intermedial frequency ranges but is too high to yield a reasonable band discrimination within our low frequency range and too low to capture the natural bandwidth of broadband high-frequency activity.

For our analysis of the relationship between pupil size and aperiodic activity we opted for the pwelch approach with a 2-sec window length instead. Longer segments reduce noise in the spectral estimation, which improves the FOOOF fit of the aperiodic part (slope) of the power spectrum. Therefore, we prioritized goodness-of-fit of the FOOOF model over temporal resolution. Critically however, this analysis was based on a feature of the whole power spectrum (which were most practically derived via pwelch rather than reconstructed from wavelet-based decomposition) and did not underlie striking a careful balance between temporal and spectral resolutions in the same way as the frequency-specific estimates in the other analyses. This analysis was run in parallel with the test for non-linear relationships, which therefore also used the pwelch approach.

We have made our rationale explicit in the Methods section as follows:

Added to Methods section ‘MEG spectral analysis (all sites)’:

“Each frequency band was defined as the halve-octave band around the respective center frequencies, thus adapting spectral and temporal widths of wavelets to accommodate wider bands and finer temporal resolution for higher frequencies.”

Added to Methods section ‘Source-level power spectra and spectral parametrization’:

“Note that we opted for a different spectral decomposition method here than the wavelet-based approach (as described above) for practical reasons: Welch's periodogram method yields a direct estimate of the spectrum, which is convenient for parameterization as this analysis focused on the 1/f feature of the power spectrum rather than on power at individual frequencies.”

#2 An additional control analysis exploring whether the spatial correlation maps in are associated with source power in the presented frequencies. For example, figure 5D shows the spatial map of the quadratic coefficient at 11 and 19Hz – how closly related is this topography to the simpler spatial distribution of 11Hz and 19Hz power?

This is an interesting point. In order to test the similarity between the maps of the pupil-power coupling (linear as well as quadratic) and the source maps of power in the corresponding frequency range, we computed the spatial correlation between the two. The findings are summarized in Author response image 4 . In short, while the spatial distribution of band-limited power overlaps partially with the pupil-MEG correlation maps, especially in the low frequency range (see panel B), it explains only a fraction of the variance (< 4%). Hence, regressing out the spatial power maps of the pupil-MEG maps has little effect on the results (see panel C).

**Author response image 4. sa2fig4:** Comparison of the spatial distribution of power and the pupil-MEG correlation maps across four frequency bands of interest. (A) Distribution of source power. Power estimates were adjusted in order to correct for the “center of the head” bias inherent to beamformer source estimates. (B) Correlation between spatial maps of (normalized) power and the spatial maps of pupil-MEG correlations (as shown in Figure 4D). Shaded area depicts the standard error of the mean and gray horizontal lines indicate correlations significantly different from zero after adjusting for the family-wise error (q=0.1). (C) Pupil-MEG correlation maps after regressing out the spatial distribution of power (separately for each participant). (D) Correlation between spatial maps of (normalized) power and ß2 (i.e., the estimated strength of the quadratic relation between pupil and MEG power).

The source maps of power explain even less variance when compared with the spatial distribution of the ß2 parameter, which estimates the strength of the quadratic relation between pupil and power fluctuations, with all correlations <0.1 (< 1% explained variance).

#3 From the discussion in Gao et al.,: “Hence, we propose that slope changes in a particular frequency region (30-70 Hz) correspond to changes in E:I balance, while making no claims about other frequency regions”. Not sure what to recommend about the frequency range difference here – at the least it should be acknowledged and discussed. Ideally this analysis should be matched to Gao et al., given how critical this reference is to the discussion and interpretation.

Thank you for pointing this out – we agree that the choice of frequency range is not sufficiently justified in our manuscript. In their paper, Gao et al., generated data by simulating a simple statistical model (their Figure 1, Author response image 5) in order to obtain a rough frequency range where PSD is sensitive to changes in excitation-inhibition ratio, which they then apply to data recorded in humans (ECoG; 30-50 Hz) and monkeys (LFPs; 40-60 Hz). Importantly, however, it is evident from their Figure 1G that PSD slope in lower frequency ranges exhibits even higher correlations with excitation-inhibition ratio (the first data point in panel 1G is centered at 20 Hz and represents the frequency range from 10 Hz to 30 Hz). In order to make this more explicit, we simulated the model of Gao et al., (using publicly available code: https://github.com/voytekresearch/EISlope/) and extended the frequency range to lower frequency bands. As Author response image 5 shows, the correlations between excitation-inhibition ratio and PSD slope are even higher for frequency ranges not shown in the paper of Gao et al., and are highest in the range chosen for the current manuscript (shown in blue).

**Author response image 5. sa2fig5:** Spearman correlation between changes in excitation-inhibition ratio and PSD slope. Shown are center frequencies (+/- 10 Hz). Gray dotted lines show the standard deviation across five model runs.

Consistently, by simulating a biologically more plausible conductance-based model comprised of leaky integrate-and-fire neurons, Trakoshis et al., (2020) demonstrated that the slope-based inference of E/I ratio is valid for both higher and lower frequencies (c.f. Figure 1D,E). In their own words: “Slopes from the low-frequency (Figure 1D) and high-frequency region (Figure 1E) increase when g is reduced (i.e. E:I ratio augmented)”, where the “low frequency region” is defined as 1-30 Hz and the “high frequency region” defined as 30 – 100 Hz. Hence, according to both, the simpler model (Gao et al.,), as well as a biologically plausible LIF model, the frequency range chosen in the current paper is justified.

In order to make this more clear in the paper, we now point to the aforementioned articles in order to better justify the frequency range chosen in our analysis:

“To this end, we parameterized the power spectra of consecutive data epochs of the source-projected MEG data in a frequency range from 3 to 40 Hz, separately for all source locations (Donoghue et al., 2020). Simulations of a biologically plausible neural network (Trakoshis et al., 2020) as well as empirical insights from optogenetic stimulations in neonatal mice (Chini et al., 2021) show that spectral slopes extracted from the chosen frequency range are in fact sensitive to changes in the underlying ratio between excitation and inhibition. The resulting parameter estimates described the periodic and the aperiodic components per epoch (data segmentation as in the previous section; see Methods).”

Additional points#4 It isn't clear where the three frequency bands of interest (line 188: 2-4Hz, 8-16Hz, and 64-128Hz) come from. I think this is based on Figure 1 but only the Glasgow site appears to show distinct clusters in these bands. Hamburg and Munster are significant across the whole 2-32Hz range and Hamburg is additionally significant from 32-128Hz. On this basis, the exclusion of the 4-8Hz and 16-64Hz range seems a bit odd. Some additional clarification and justification would be helpful – apologies if I have missed this in another section.

Note first that the 4-8Hz and 16-64 Hz ranges were not generally excluded. Instead, they were left out for reasons laid out below and for the cross-correlation analysis only (Figure 3). Other analyses included these bands.

The three frequency bands, referred to by the reviewer, are a result of their characteristic morphologies in the cross-correlation analyses (Figure 3), and not a result of analyses depicted in Figure 2 (which, we assume, the reviewer refers to as ‘Figure 1’). For the banding in the cross-correlation results (Figure 3), we took a careful approach and excluded bordering frequencies to emphasise the characteristic pattern for each band.

In general, our approach was to be as minimally prescriptive as possible with respect to “banding”. We are using it descriptively where the data commends it and as a means of data reduction to ease illustration and interpretation – see e.g. source reconstructions that separate bands more “classically” into theta, α, β and γ ranges. Further justifications are given by the typical subdivision into low- and high-frequency bands in rodent research.

Figure-2 findings are based on mutual information between pupil and power envelopes, which is sensitive to monotonic and non-monotonic relationships in the data, and can therefore be seen as a blanket approach to show that (1) there are relationships to be expected in the data across virtually all frequencies in follow-on, fine-grained analyses, and (2) this is widely consistent across labs. We find that this is captured in the following section of the Results section but will amend if the Reviewer thinks additional clarification is necessary:

“For all three MEG laboratories, standardised-MI spectra showed similar patterns of widespread pupil-brain associations across frequencies with prominent peaks in the 8-32 Hz range (Figure 2A). MI scalp distributions within this frequency range further underpinned the commonalities between recording sites despite different MEG systems and setups (Figure 2B). Significant relationships with pupil **size** were also evident for the low-frequency (4-8 Hz) and high-frequency (64-128 Hz) components of MEG-power (Figure 2A) in all three datasets.”

#5 related to 4, subjectively the MI spectra across the three sites don't seem to be all that similar, a stronger justification for combining subjects in subsequent analyses would be reassuring. Given that this is the jumping off point for the rest of the paper it would be good to revisit this figure to emphasise the points of similarity – the supporting spectra in supplemental materials and topos show between site similarity more convincingly.To formalise this, can you repeat the between subject correlation matrix from figure S1B for the MI spectra and see if there is substantial banding per-site? if site-specific bands are visible here, then a mixed-modelling approach or the inclusion of a site-covariate in subsequent analyses would be desirable.

As pointed out in response to point 4 above, the crucial take-away from Figure 2 should be that there are systematic pupil-brain relationships across all (or most) frequencies for all three datasets. As mentioned by the Reviewer, pooling of the data was justified by a range of similarities, also taking into account the results of the cross-correlation analysis shown in Figure 3A and Figure 1—figure supplements 1 and 2.

Additionally, we have revised the first paragraph of Results section to point out analyses relating to Figures S1 and S2 (in the revised manuscript referred to as Figure 1—figure supplements 1 and 2), and their purpose in justifying a joint analysis of all three datasets more strongly:

“For each of the three MEG laboratories, we obtained time courses of MEG activity and of pupil diameter fluctuations. A first control analysis of the recordings (Figure 1—figure supplements 1 and 2) established that they were highly comparable between laboratories, despite the differences in locations, setups, and participants.”

We have further revised Figure S2 (revised version: Figure 1—figure supplement 2) to include more information and adapt the design of S1 (revised version: Figure 1—figure supplement 1).

For this response letter, we have also produced the requested correlation plot for individual spectra of standardised Mutual information (see Author response image 6 ). Note first that this similarity measure is necessarily noisier than related plots based on MEG or pupil power spectra (see, e.g., new plot Figure 1—figure supplement 2) due to relatively small effect sizes (e.g., correlations in the range of [-0.05; 0.05], see Figures 3 and 4). Although there seems to be some clustering around the Hamburg data, this is not exclusively lab-specific and is most likely explained by different signal-to-noise ratio between the labs that are due to the difference in the lengths of the resting-state blocks (Münster = 5 min, Glasgow = 7 min, Hamburg = 10 min; as reported in the Methods section), giving the Hamburg data the highest SNR. We assume that including a factor of “recording lab”, if explaining systematic variation in the data, would primarily indicate this SNR effect, and therefore would be uninformative.

**Author response image 6. sa2fig6:** Participant-by-participant correlation of standardised Mutual Information (MI) spectra. Colour coding along x and y-axes indicate the recording laboratories.